# Ultrafast motion in a third generation photomolecular motor

**Palas Roy**[1,3], **Wesley R. Browne** [2], **Ben L. Feringa** [2] ✉ **& Stephen R. Meech** [1] ✉

Controlling molecular translation at the nanoscale is a key objective for development of synthetic molecular machines. Recently developed third generation photochemically driven molecular motors (3GMs), comprising pairs of overcrowded alkenes capable of cooperative unidirectional rotation offer the possibility of converting light energy into translational motion. Further development of 3GMs demands detailed understanding of their excited state dynamics. Here we use time-resolved absorption and emission to track population and coherence dynamics in a 3GM. Femtosecond stimulated Raman reveals real-time structural dynamics as the excited state evolves from a Franck-Condon bright-state through weakly-emissive dark-state to the metastable product, yielding new insight into the reaction coordinate. Solvent polarity modifies the photoconversion efficiency suggesting charge transfer character in the dark-state. The enhanced quantum yield correlates with suppression of a low-frequency flapping motion in the excited state. This detailed characterization facilitates development of 3GMs, suggesting exploitation of medium and substituent effects to modulate motor efficiency.

Biology utilizes ATP driven motors to efficiently transport macromolecules along cytoskeletal filaments[1]. The synthesis of analogous artificial molecular motors capable of controllable translational motion has long been an objective for chemistry[2–4]. Over the past two decades, light-driven molecular motors that exploit cis-trans photoisomerization around an ethylenic double bond axle have attracted wide interest[5–7]. Unidirectional rotation in such motors was achieved by introducing steric constraints and stereogenic centres to create helical shapes with asymmetric barriers. The first generation of unidirectional motors (see Fig. 1a) featured two equal halves bearing two stereogenic centres to provide helical chirality[8]. Subsequently, second-generation motors consisting of two distinct halves (rotor and stator) with only one stereocenter were developed, and shown to operate at up to MHz frequencies[9–11]. Both generations first undergo photoisomerization to form a metastable product that relaxes via thermal helix inversion to form a stable product, completing a 180⁰ rotation. These motors, therefore, offer a means of converting light energy into

directional rotational motions, but early efforts to develop light driven motors for translational motion required a significantly more complex architecture[12].

The desired efficient conversion of light-driven unidirectional rotational motion into unidirectional translation can potentially be achieved in the recently described third-generation motors (3GMs) comprising two rotors, two axles, and a pseudoasymmetric center (Fig. 1a)[13–15]. These 3GMs are a combination of two second-generation motors possessing opposite helicity with respect to the bridging unit. On irradiation with blue light, one rotor isomerizes (around axle-1) to form a metastable product, which relaxes to a stable product via thermal helix inversion, completing the first rotation. Subsequently, the second rotor is excited, isomerizes and rotates (around axle-2) in the same direction to complete the cycle (Fig. 1b). Uniquely, these light-driven motions in 3GMs suggest the possibility of molecular scale translational crawling along a suitable substrate. Thus, 3GMs are a promising development in artificial light-driven transport. However,

[1]School of Chemistry, University of East Anglia, Norwich Research Park, Norwich NR4 7TJ, United Kingdom. [2]Stratingh Institute for Chemistry, University of Groningen, Nijenborgh 4, 9747AG Groningen, The Netherlands. [3]Present address: School of Basic Sciences, Indian Institute of Technology Bhubaneswar, Argul, Odisha 752050, India. ✉e-mail: b.l.feringa@rug.nl; s.meech@uea.ac.uk

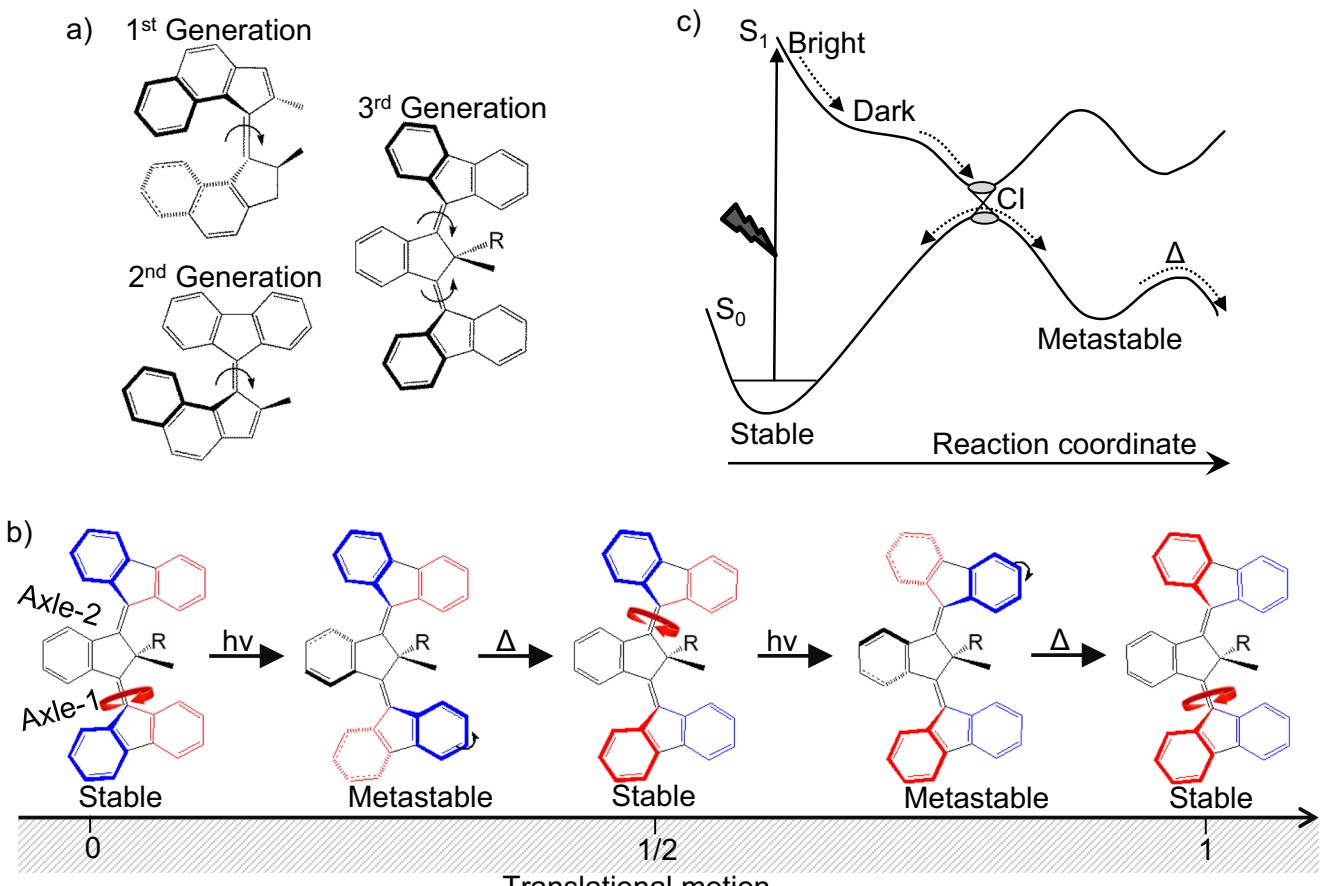

**Fig. 1 | Structures of successive generations of unidirectional motors and the unique function of 3GMs. a** Structures of three generations of unidirectional light-driven rotational motors; thicker lines indicate atoms above the plane of the page, light lines below. **b** Illustration of the potential of cooperative rotation in 3GMs to support translational motion following sequential photochemical (*hv*) and thermal (Δ) steps localized on axle-1 then axle-2. Colors are a guide to the eye for the motions of the otherwise symmetric rotors (structures from Kistemaker et al.[14]). **c** Potential energy diagram showing photochemistry driven by light and heat.

major challenges remain in enhancing the photochemical yield and speed of rotation. Guidelines for designing more efficient 3GMs will require a detailed understanding of excited state dynamics; this is the topic of the present work.

Ultrafast photodynamics of first- and second-generation motors have been extensively studied[16–28]. Photoexcitation populates a Franck Condon 'bright' state that is emissive in nature. Driven by strong steric repulsion, the bright-state undergoes relaxation within ca 100 fs to form a much less-emissive dark-state intermediate. Thereafter, the molecule passes through a conical intersection (CI) to recover the ground state in 1–10 ps, bifurcating to populate either the metastable product or the initial state (Fig. 1c). Thus, the nature of the excited state intermediate is critical in determining the yield of metastable product. The photochemistry of this state can be significantly modified by substituents[24,29]. However, the real-time structural evolution and how to tune the excited state potential energy surface to optimize quantum yield are not yet well-understood. Here we use a combination of femtosecond electronic and vibrational spectroscopic tools to study a third-generation motor with a phenyl substituent at the pseudochiral centre (3GMph) (Fig. 2a inset). Ultrafast fluorescence and transient absorption characterize population relaxation and coherent vibrational dynamics in 3GMph. Femtosecond stimulated Raman spectroscopy (FSRS) adds structural dynamics data at the level of vibrational spectroscopy, with frequency assignments being supported by density functional theory calculations.

## Results

### Electronic spectroscopy

The steady-state electronic spectra of 3GMph in acetonitrile (ACN) are shown in Fig. 2a. The absorption maximum is at 437 nm while the emission is weak and broad, peaking at 580 nm. These transitions are significantly red-shifted from the corresponding second-generation motor, indicating a change in electronic structure when two motors are coupled. The fluorescence is weak, with a quantum yield of $<3 \times 10^{-5}$ (Supplementary Note 3). To understand the origin of this weak emission, time-resolved fluorescence up-conversion measurements (TRUC) were performed (Fig. 2b). Bi-exponential fitting of the fluorescence decay at 560 nm shows the decay is dominated by a fast component of $140 \pm 20$ fs, with a minor (11%) second component with a $1.6 \pm 0.4$ ps decay time. Residuals recovered from the fitting (bottom panel in Fig. 2b) show oscillations due to coherently excited Raman active modes in the electronically excited state[30]. Such ultrafast non-single exponential fluorescence was previously observed in both first and second generation motors, and indicates ultrafast decay of the bright Franck Condon state to a 'dark' excited state – i.e. one with a significantly reduced transition moment – which has a picosecond lifetime[16,17]. The combination of ultrafast bright-state relaxation, decreased transition moment and picosecond dark-state decay leads to the low fluorescence yield. TRUC measurements were performed at a range of emission wavelengths (see Fig. 2b inset). The amplitude of the picosecond component increases from 490 nm to 620 nm (Supplementary Table 1) showing that the short-lived bright Franck Condon

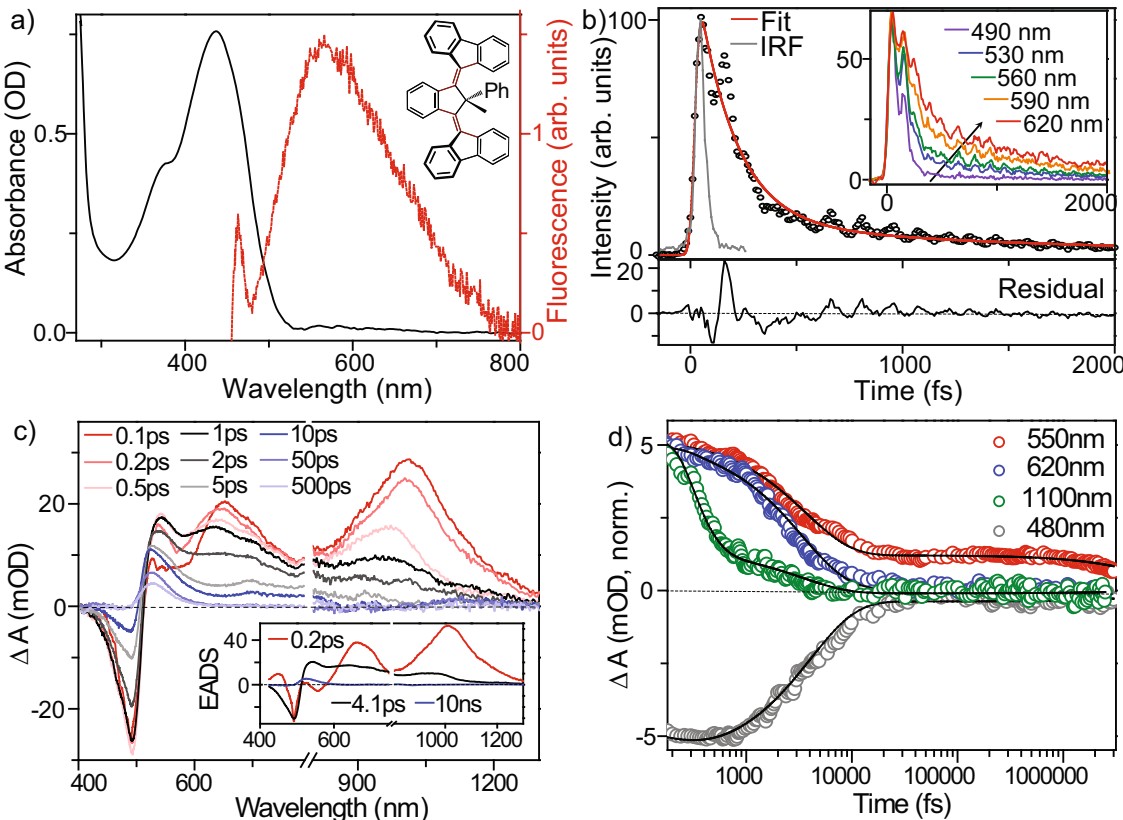

**Fig. 2 | Electronic spectra and excited state dynamics of 3GMph. a** Steady-state absorption (black) and emission (red) spectra of 3GMph in ACN. Inset: chemical structure of 3GMph. **b** Time-resolved fluorescence measured at 560 nm after excitation at 410 nm. The red line shows the fit to a biexponential function and the bottom panel is the residual, which is oscillatory, indicating a role for coherently excited vibrational modes in excited state dynamics. The grey line shows the instrument response (width 43 fs). Inset: fluorescence dynamics at five emission wavelengths; **c** Transient vis-NIR absorption spectra at different time delays

showing spectral evolution in excited 3GMph; spectra were measured in two spectral windows and stitched together at ca 800 nm. Inset: components of the transient absorption data recovered from global analysis assuming a simple sequential kinetic model with a single intermediate state to yield three evolution-associated difference spectra (EADS); **d** Population dynamics at four different wavelengths (circles) with quality of fit of the sequential global analysis model shown (solid line).

state is blue-shifted compared to the dark-state. In addition, the wavepacket modulation in the emission kinetics is preserved at all wavelengths and the coherences remain in-phase (Fig. 2b inset). This is consistent with the coherently excited vibrational mode modulating the transition moment for emission, rather than the transition energy (which would result in out-of-phase oscillations on the red and blue edges). This suggests a coordinate dependence of the transition moment (a non-Condon effect). This result, which has been reported in other excited state isomerization reactions, is consistent with these low frequency modes playing a role in driving the bright to dark-state evolution[26,31].

To picture the complete photodynamics, we performed transient absorption measurements (TA) on 3GMph in ACN and cyclohexane. Figure 2c shows TA spectra in ACN at different time delays. Negative signals correspond to ground state bleach (GSB) or stimulated emission (SE) while positive signals indicate excited state absorption (ESA) or metastable product state formation. The spectrum at 100 fs has a broad ESA with peaks at 550, 670 and 1000 nm. This spectrum evolves within 1 ps to yield a new ESA spectrum with peaks at 560, 720 and 950 nm. This spectrum further evolves in picoseconds to yield a feature at 530 nm which persists for >500 ps, and does not undergo further spectral evolution. Global analysis reveals three states connected in a sequential scheme: bright Franck-Condon state → dark-state → metastable product (see Fig. 2c inset). Fitting of the kinetics of the GSB (at 480 nm) and ESA (at 550, 620, 1100 nm) using this simple single intermediate model are shown in Fig. 2d. The fit is good, and can only

be improved by adding additional intermediates, which yield no new spectral information; such an improvement can be assigned to a degree of inhomogeneity in the dark-state decay, as previously observed for first and second generation motors[16,18,24]. The same good fit of this simple model is found in all solvents studied and applies equally to the FSRS data (below). The dominant decaying component $(200 \pm 40\ fs)$ in the TA matches the ultrafast emission decay time (Fig. 2b, Supplementary Table 1) and is thus assigned to decay of the bright Franck-Condon state. During this sub-picosecond decay the GSB (480 nm) does not recover at all, while the SE at 560 nm does decay (Fig. 2c,d). This result is compelling evidence that the ultrafast decay is indeed a bright-state to dark-state reaction occurring on the excited state potential energy surface prior to internal conversion.

The picosecond component $(4.1 \pm 0.5\ ps)$ from the global analysis is assigned to the dark-state decay. This is longer than the $1.6 \pm 0.4\ ps$ decay recovered from TRUC. The difference is outside experimental error and becomes significantly larger in nonpolar cyclohexane (see below). This is consistent with the inhomogeneous dark-state decay noted above, with shorter-lived components contributing more to the emission through a larger transition moment[18]. The GSB also begins to recover on the picosecond timescale indicating repopulation of the stable ground state; its recovery is incomplete even at long times because of formation of the metastable product. The metastable photoproduct absorbs at >450 nm, red-shifted from the stable ground state absorption (Fig. 2a,c). This is consistent with previously reported steady state difference spectra for other 3GMs at reduced temperature

(see Supplementary Fig. 7)[14], and shows that the TA has captured the full evolution from FC to metastable product state, yielding the most complete characterization yet of excited state dynamics in photo-molecular motors.

## Ultrafast Raman spectroscopy

To investigate structural dynamics, we apply resonant FSRS to recover transient vibrational spectra, which can provide insight into intermediate state structures. FSRS is a multipulse coherent Raman experiment in which an excitation pulse (pump) initiates the excited state reaction, and the stimulated Raman spectrum of intermediates is measured by combination of a narrowband picosecond Raman-pump pulse with a broadband femtosecond source. This yields the resonance Raman spectrum of excited state species resonant with the Raman-pump pulse. The experiment is described in detail elsewhere[32,33].

Figure 3a shows the ground state stimulated Raman spectrum (GSR) of the stable form of 3GMph, which is dominated by a signal at $1553\,cm^{-1}$. To assign this, a DFT calculation of the vibrational spectrum was performed; the $1200$-$1600\,cm^{-1}$ region is plotted in Fig. 3b (see Supplementary Fig. 17 for the full spectrum). Based on these data, the $1553\,cm^{-1}$ signal is assigned to contributions from a pair of modes calculated at $1540$ and $1554\,cm^{-1}$ largely localized on the ethylenic axles, and arising from asymmetric and symmetric $C=C$ stretching respectively (Supplementary Fig. 18). The $1100$-$1500\,cm^{-1}$ region is assigned through DFT to ring stretching plus C-H bending modes. The low-frequency modes in the stable form GSR spectrum are all weak (<8% of the amplitude of the $1553\,cm^{-1}$).

The experimentally recovered FSRS at 200 fs after excitation, measured in resonance with the excited state absorption at 560 nm, is marked as $ESR_{560nm}$ in Fig. 3a (Supplementary Fig. 11 for details of baseline correction procedure and Supplementary Fig. 23 for details of the subtraction of the non-resonant solvent and pre-resonant GSR contributions). The major feature is a complex lineshape around $1570\,cm^{-1}$; in addition, some low-frequency modes are enhanced compared to the GSR, especially below $1000\,cm^{-1}$. The complex lineshape includes a differential feature centered around $1555\,cm^{-1}$, close to the main GSR feature. Such complex FSRS lineshapes have been discussed in detail elsewhere, and cataloged by Weigel et al.[34]. In addition to the expected positive stimulated Raman signal enhanced by the $S_1$ to $S_n$ transient absorption, contributions from $S_1$ to $S_0$ stimulated emission (variously described as Raman induced nonlinear emission, RINE, or inverse Raman) and $S_0$ to $S_1$ pre-resonance Raman all lead to negatively signed contributions in the difference data at or near the wavenumber of GSR modes[35,36]. These two separate contributions lead to distinct signals in addition to the positive $S_1$ Raman spectrum sought, one (RINE) characterized by a differential lineshape and one ($S_0$-$S_1$ bleach feature) by a negative contribution from the ground state Raman. Mathies and coworkers reported that tuning the Raman-pump away from the SE eliminated negative $S_1$ to $S_0$ features[37,38]. Thus, FSRS measurements with the Raman-pump at 650 nm were made, where SE is weaker but $S_1$ to $S_n$ resonance is retained (Fig. 2a). $ESR_{650}$ data show a combination of sharp and broad positive bands in the $1500$-$1700\,cm^{-1}$ region (Supplementary Fig. 10c), with the negative feature largely suppressed, although its contribution is still resolvable in the long-time data as a weak negative feature (compare Supplementary Figs. 9 and 10). An even better experiment would be use of a Raman pump beyond 1 micron (where the broad SE is negligible, Fig. 2a) but current detectors provide insufficient signal-to-noise. Hence, we have been unable to measure with a Raman pump wavelength >700 nm, and those data did not show a RINE feature at early time (Supplementary Fig. 21), which is essentially identical to the 650 nm data. In addition, we observed that the GSR signal was also reduced four-fold at 650 nm compared to 560 nm, as expected when moving the Raman-pump further to the red in the pre-resonance Raman regime. Two features of these data suggest that the dominant

contribution to the negative amplitude is the bleached GSR signal. First, the negative feature persists at all delay times, whereas the RINE features usually decay on the sub-picosecond timescale, faster than the population decay[36,39]. Secondly, even though the negative feature is well resolved at long times (especially with the 560 nm Raman pump) there is no sign of the positive lobe invariably associated with the differential lineshape from RINE. If we assume the dominant contribution is bleached GSR, the negative contribution can be corrected by quantitative bleach filling, i.e. adding controlled amounts of the measured GSR to the $ESR_{560}$ data, as has been described elsewhere[40–42]. Initially, we fill the 760 ps data, where only metastable product and bleach contribute, to recover a smooth metastable product Raman spectrum (Supplementary Fig. 12a). We then add to all $ESR_{560}$ data a fraction of the GSR determined by the bleach population amplitude recovered from TA (Fig. 2d and Supplementary Fig. 12b). Thus, the bleach-filled excited state Raman data in Fig. 3c, d are obtained; equivalent data were recovered from $ESR_{650}$ (Supplementary Figs. 13, 15). This successful quantitative application of the bleach filling model points to the GSR bleach being the main factor distorting the lineshape of the $S_1$ Raman spectrum, which has then allowed us to isolate the desired spectrum (Fig. 3c, d). The dominance of the GSR contribution at $1555\,cm^{-1}$ is consistent with the observation of an intense GSR signal at this frequency. However, although we do not require any additional RINE contributions to fit the present lineshape we cannot rule out the possibility of a contribution at early times, as discussed below.

Sub-picosecond bleach-filled FSRS (Fig. 3c) show Raman active modes of the bright-state in low and high frequency regions. The dominant high-frequency contribution ($1500$–$1600\,cm^{-1}$) has an asymmetric lineshape, which is well fit by a pair of Gaussians (fitting parameters, Supplementary Table 2). Based on ground state DFT, these two modes can be assigned to axle $C=C$ stretching (the ground state mode displacements at 1554 and $1540\,cm^{-1}$ are shown in Supplementary Fig. 18). The appearance of two distinct modes is assigned to electronic excitation localized on one axle causing an increased splitting of the pair of axle modes within the same motor, resulting in the bimodal narrow ($1565\,cm^{-1}$) plus broad ($1607\,cm^{-1}$) lineshape observed. Although two modes are predicted in this $C=C$ stretch region we cannot completely rule out that the distinctive lineshape observed has contributions from uncorrected RINE features (although the facts that the amplitude of bleach filling required was quantitatively predicted by the population from TA, and that the 650 and 700 nm raw FSRS data show no hint of a differential lineshape at early times, yet the spectrum recovered at <1 ps is identical to that at 560 nm, see Supplementary Fig. 21, both point to a dominant bleach filling mechanism). In the case that RINE does contribute to the lineshape, it would simply be one broad band and the exact frequencies of the two modes less certain. What is clear is that both modes are blue-shifted compared to GSR. We assign this to excitation localized on one axle decoupling the two halves of the motor, breaking the extended conjugation. This enhances the $C=C$ bond order, giving rise to the observed blue-shift. In addition to the bridge $C=C$ contribution, there is enhanced activity in the low-frequency Raman spectrum at 350, 490, and $600\,cm^{-1}$ of the bright-state compared to the stable ground state (Fig. 3c). Ring and CH modes in the $1100$–$1400\,cm^{-1}$ region are also enhanced.

The FSRS temporal evolution (Fig. 3c) shows that the bimodal lineshape decays on the subpicosecond time scale leaving a weaker mode at $1539\,cm^{-1}$ seen fully formed in the 2 ps spectrum. In $ESR_{560}$ this single mode broadens and shifts to the red with time (i.e. the $C=C$ bond strength decreases during excited state evolution to the dark state and prior to the formation of the ground state product). Peak area dynamics of the $1000$–$1750\,cm^{-1}$ region were fit to yield relaxation times of 180 fs and 6 ps (plus a constant). These rates are consistent with TA data (above and Supplementary Fig. 12c) indicating that the

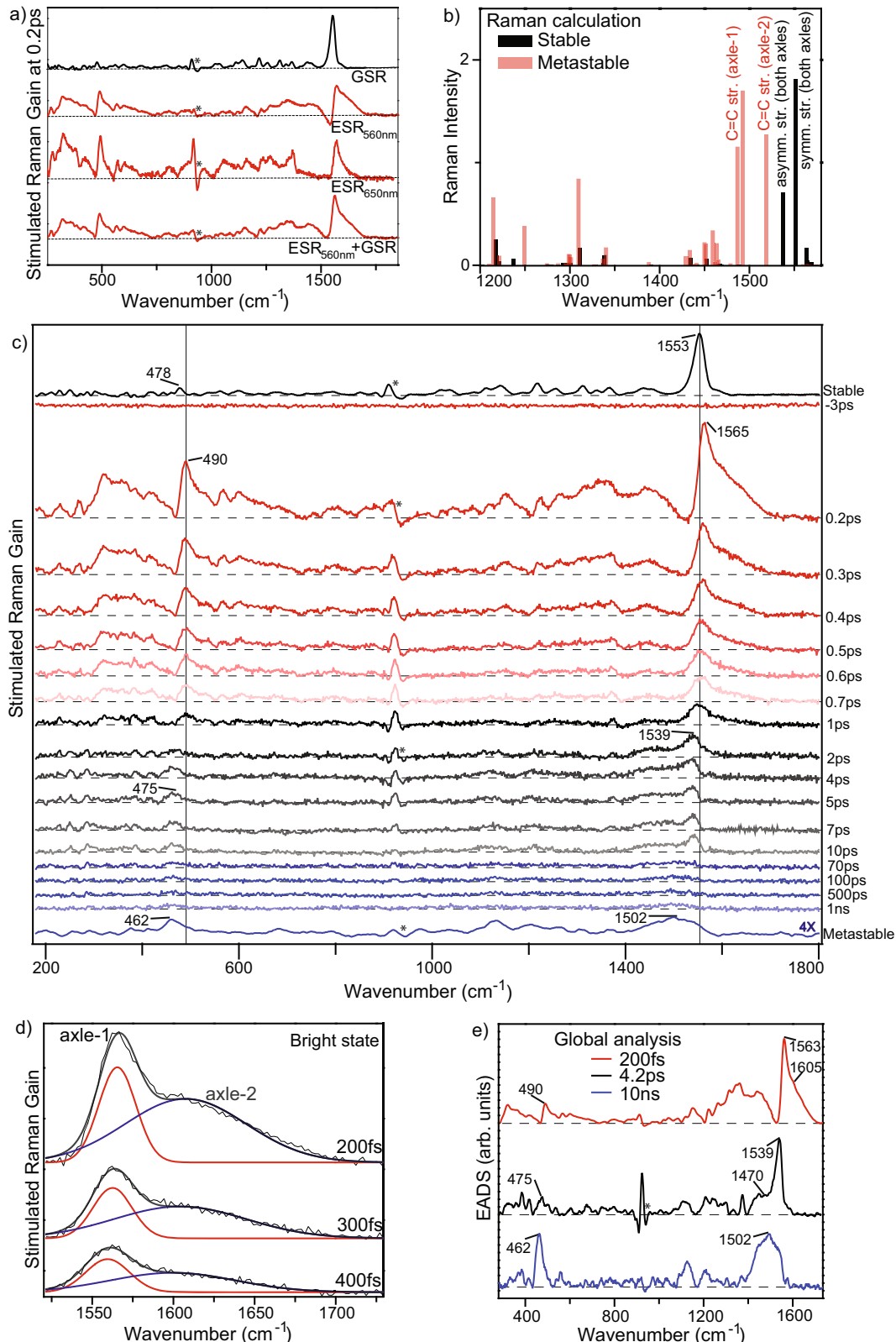

**Fig. 3 | Characterization of 3GMph excited state structural dynamics by transient Raman. a** Measured ground state Raman (GSR, pump off) and excited state Raman (ESR, pump on) spectra of 3GMph, the latter at two different Raman-pump wavelengths, 560 nm and 650 nm, both recorded 200 fs after excitation in ACN. The true ESR spectrum at 560 nm has been recovered from the complex lineshape by bleach filling (i.e. ESR$_{560nm}$ + GSR, See text). Pump pulse at 440 nm; *indicates solvent artifact. **b** DFT calculated Raman spectra of the stable (black) and metastable (red) forms of the ground state. The computed frequencies were scaled by 0.98[43]. **c** The bleach filled ESR$_{560nm}$ spectra at different pump-probe delays (pump 440 nm, Raman-pump at 560 nm; the 650 nm Raman pump data and unfilled data sets are shown in Supplementary Fig. 10); *indicates solvent artifact. **d** Early time evolution of the bleach filled ESR$_{560nm}$ in the bimodal ethylenic stretch region (1520–1750 cm$^{-1}$), here fit by two Gaussians (red at 1565 cm$^{-1}$ and blue at 1607 cm$^{-1}$). **e** Global fitting components of the FSRS data in (**c**) assuming the same three-state sequential model as for TA with the same fixed time constants.

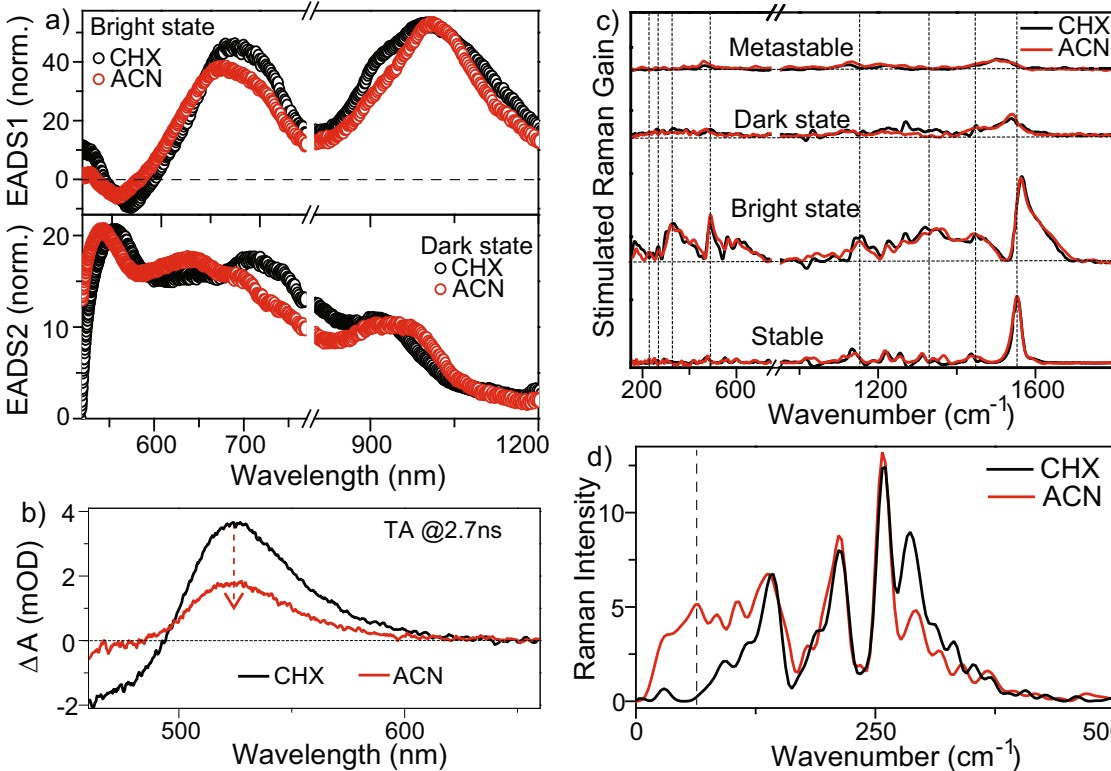

**Fig. 4 | Solvent polarity dependence to the 3GMph dynamics. a** Evolution associated difference spectra of bright (top, EADS1) and dark (bottom, EADS2) states **b** nanosecond absorption spectra of metastable product, which decreases in amplitude between cyclohexane (CHX) and acetonitrile (ACN) solvents

**c** femtosecond stimulated Raman spectra (bleach filled) of stable, metastable, bright and dark-states **d** impulsive Raman spectra obtained via Fourier transformation of the oscillatory residual from time-resolved fluorescence measurements.

FSRS amplitude correlates with bright to dark-state evolution. To resolve Raman spectra of the distinct states we performed the global analysis with fixed time constants taken from TA data (0.2 ps, 4.1 ps and 10 ns). The 4.1 ps time constant from TA global analysis is assumed to be more accurate than the 6 ps from fitting the FSRS data, as the TA data have better signal-to-noise and were measured at a larger number of time delays in the picosecond time range. An arbitrary long time of 10 ns is selected to capture the final long-lived metastable state contribution. This global analysis recovers evolution associated Raman spectra of the bright, dark and product states respectively (Fig. 3e). The subpicosecond decrease in peak area in the axle region suggests bridge C = C modes are of lower amplitude in the dark- than in the bright-state, reflecting either weaker resonance enhancement or an intrinsically lower Raman cross section in the dark-state. The bimodal bright-state spectrum (Fig. 3d) relaxes to the dark-state, which has a well-defined Raman active mode at 1539 cm$^{-1}$ and an additional broad feature around 1490 cm$^{-1}$ (see above and Fig. 3e). The strong low-frequency modes observed in the bright-state persist in the dark-state, although the mode at 490 cm$^{-1}$ is red-shifted by 15 cm$^{-1}$ (Fig. 3e). The ring modes in the region 1100–1400 cm$^{-1}$ are less prominent in the dark-state.

The final evolution associated Raman spectrum (Fig. 3e) is assigned to the metastable product, the only long-lived state seen in TA. Its contribution is stronger in ESR$_{560}$ than ESR$_{650}$ due to stronger resonance enhancement. This product has a broad Raman signal at 1502 cm$^{-1}$ (Fig. 3c,e). Based on the DFT calculation for the metastable ground state (Fig. 3b) this is assigned to two overlapped independent C = C stretching modes calculated at 1490 and 1516 cm$^{-1}$ (Supplementary Fig. 19). This is different to the stable form, which displayed coupled symmetric and anti-symmetric C = C stretching vibrations (Supplementary Fig. 18). The 55 cm$^{-1}$ red-shift is predicted in DFT,

where the C = C stretch mode for axle-2 is now localized, while that for axle-1 is delocalized over the phenyl ring (Fig. 3b and Supplementary Fig. 19). The metastable product also has enhanced Raman activity at low frequency compared to the stable form, most notably in a 462 cm$^{-1}$ mode assigned through DFT to CH out-of-plane bending (Supplementary Fig. 19). This mode is 16 cm$^{-1}$ red-shifted compared to the corresponding stable state form, which mirrors the red-shift seen in the bright- to dark-state evolution of the 490 cm$^{-1}$ mode (above).

## Polarity effects

To investigate the effect of solvent polarity on dynamics in 3GMph, we recorded ultrafast electronic and vibrational spectra in nonpolar cyclohexane (CHX) and compared them with polar ACN data (Supplementary Fig. 1a). The absorption spectra are essentially independent of solvent polarity, while the fluorescence in CHX is slightly blue shifted compared to ACN. The small decrease in Stokes loss compared to polar ACN suggests a more polar character in the emissive state compared to the ground state. In the time domain, the TRUC at 560 nm in CHX shows a dominant ultrafast decay of 125 ± 10 fs along with a minor (12%) 1.5 ± 0.4 ps component (Supplementary Fig. 4a and Supplementary Table 1). Thus, within experimental error, there is no solvent polarity effect on fluorescence decay kinetics.

TA in nonpolar CHX are shown in Supplementary Fig. 6. The evolution is qualitatively similar to that in ACN, but with different time constants. Global analysis again shows three components with a single intermediate (Supplementary Fig. 8) with time constant of 0.17 ps for bright-state decay to a 12.2 ps lifetime dark-state, which decays to the long-lived (>10 ns) metastable product. Therefore, decreased polarity of CHX does not alter the bright-state lifetime, but extends the dark-state lifetime by a factor of three. Note that the marked difference in the long components recovered from TRUC and TA for CHX, is further

evidence (see above) for a distribution of dark-state structures with distinct radiative transition moments.

In Fig. 4a the ESA of dark- and bright-states obtained from global analysis in the two solvents are compared. The bright-state spectra are similar, although in ACN the ESA at 1000 nm is narrower and the 675 nm peak slightly blue-shifted (by 10 nm) compared to CHX. The dark-state ESA (Fig. 4a) show a stronger dependence on solvent polarity, with ACN inducing a 70 nm blue-shift in the peak at 710 nm along with a broadening of the feature at 920 nm. This suggests that dark-state energy levels are modified by solvent polarity, consistent with it having a polar character. An important question is whether this solvent polarity effect on dark-state lifetime and spectra also modifies the metastable product yield. To probe this, we recorded TA spectra at 2.7 ns under identical experimental conditions of sample concentration, pump power, and pump-probe overlap (Fig. 4b). The metastable product absorption at 530 nm is doubled while the GSB is correspondingly increased. Since only the metastable and stable states contribute to the difference spectrum at long times this solvent dependence can arise in only two ways; either the metastable state is solvent dependent and is shifted under the solvent independent (Supplementary Fig. 1a) stable state spectrum, or the yield of the metastable state from the dark state is solvent dependent. To resolve this, the long-time difference spectrum was measured in a range of solvents of different polarity (Supplementary Fig. 20c). The shape and peak wavelength of the difference spectra are independent of solvent, while their amplitude is a strong function of solvent polarity, decreasing by a factor of 2 between CHX and ACN (Fig. 4) and by a factor of 15 between CHX and dimethyl sulfoxide (DMSO), Supplementary Fig. 20c. The conversion from FC through dark state to the metastable form requires an isomerization reaction, which may displace a significant solvent volume and so be sensitive to viscosity as well as polarity. The data in Supplementary Fig. 20c show that polarity is the determining factor. For example, the viscosities of nonpolar CHX and tetradecane differ by a factor of 5 but when measured under identical conditions the metastable state yield is identical. In contrast, DMSO and tetradecane differ in viscosity by less than 10% but the metastable yield is reduced by a factor of 15 in polar DMSO. This demonstrates that the formation of the metastable state is suppressed in polar solvents, from which it is concluded that medium polarity is a potential method of modulating the yield of metastable state formation in 3GMph.

The effect of solvent polarity on excited state vibrational structure was investigated; FSRS of 3GMph in CHX are compared with ACN data (Fig. 4c; see Supplementary Figs. 9 and 14 for more details). There is no major solvent dependence of the GSR spectra of either stable or metastable forms. The bright- and dark-state ESR in the two solvents are also very similar. Further, the FSRS data recorded in polar DMSO also show very similar spectral behavior although with a weak signal from the metastable state (Supplementary Fig. 22). In contrast, the very low-frequency ($<$300 cm$^{-1}$) region of the excited state Raman spectrum, probed through Fourier transform of the time domain coherent oscillation recovered from TRUC (Fig. 2b) show a remarkable solvent dependence (Fig. 4d). In both solvents, these coherent excited state modes persist in the dark-state, suggesting that they are impulsively excited in the FC state and are conserved through evolution along the excited state potential energy surface (PES) [17,30]. In CHX, Fourier transform analysis reveals modes at 146, 213, 259, and 290 cm$^{-1}$ (Fig. 4d). However, in ACN an additional mode at 53 cm$^{-1}$ is detected. This mode is consistently present at all emission wavelengths in ACN and is recovered from both time and frequency domain analyses (Supplementary Figs. 4 and 5). Based on the results of ground state DFT this Raman active mode is associated with a flapping-like motion of the separate rotors in 3GMph (Supplementary Fig. 18). Its appearance selectively in ACN suggests displacement along that coordinate is associated with the polar character of the excited state, which

evidently also enhances a radiationless decay pathway that favors the stable rather than metastable ground state (Fig. 4b). On the basis of these data we conclude that modulation of either the polar character or the low-frequency modes in 3GMs can influence the photo-isomerization yield. This important role for polarity of the dark state intermediate suggests that 3GM derivatives with electron donating or withdrawing substituents on either the fluorene or phenyl ring should be a route to controlling the photochemical yield.

## Discussion

Excited state dynamics of 3GMph were studied in detail. Photo-excitation populates a bright-state, which relaxes in $<$300 fs to generate a weakly emissive dark-state. Subsequently the dark-state decays to either the metastable product or the original ground state, with metastable state formation being enhanced in nonpolar solvents. Solvent polarity dependence of the excited state absorption highlights the role of charge separation in excited state intermediates. The solvent-dependent partitioning between metastable product and initial ground states arises from polar solvent quenching of the dark-state, enhanced by a flapping like low-frequency mode which could potentially be tuned to modulate motor performance. Similarly, synthesis of 3GMs which modify the charge transfer character of the dark-state can play a role in modifying motor efficiency.

FSRS measurements provide details on the structural evolution in the 3GM photocycle (at the level of vibrational spectroscopy). We observed stimulated Raman spectra of the bright state which were missing in previous studies on the 1$^{st}$ and 2$^{nd}$ generation motors. The sub 300 fs lifetime bright-state in 3GMph exhibits a pair of C = C modes which have a greater splitting and are shifted to higher wavenumber than in the ground state. This state decays to the dark-state, which is characterized by a red-shifted C = C stretch compared to both the initial ground state and the bright-state. The spectrum shifts further to the red and broadens on formation of the metastable product. The simultaneous observation of complementary population and structural dynamics provides the most detailed picture yet of excited processes in photomolecular rotors. These results provide the basis for future theoretical investigation and the development of experimental design principles for new motors.

## Methods

### Synthesis of 3GMph
The synthesis of 3GMph was reported previously [13,14], and is outlined in Supplementary Note 1.

### Time resolved fluorescence
Time resolved fluorescence measurements were made using the up-conversion method. The sample was excited at 410 nm using the second harmonic of a Coherent Micra 10 Ti:Sapphire oscillator. The oscillator provided 920 mW output of 20 fs 820 nm pulses at 76 MHz repetition rate. The 5 mW 410 nm excitation beam was focused to a 200-micron diameter spot on to the sample and the resulting fluorescence was collected with a reflective microscope objective. The fluorescence was focused into a nonlinear crystal where it was mixed with the residual 820 nm fundamental to generate the sum frequency signal between 307 and 353 nm depending on the fluorescence wavelength of interest. The sum-frequency was detected by a monochromator, low-noise photomultiplier and photon counter combination. The fluorescence response was time-resolved by scanning the time delay between the 410 nm excitation and the 820 nm fundamental pulses using an optical delay line. The time resolution was measured as 43 fs from the up-conversion of the solvent Raman signal. The buildup of any metastable product was avoided by flowing a 150 ml volume of the sample through a 1 mm pathlength flow cell. Further details are presented in Supplementary Note 4.

## Transient absorption

TA measurements were made using the output of a 1 kHz amplified Ti:sapphire laser producing 120 fs pulses at 800 nm of 5 mJ energy at a 1 kHz repetition rate. This source was split in three to provide an 800 nm output and to pump two optical parametric amplifiers (OPAs). The TA measurement is a pump-probe method. One OPA generated 80 fs pump pulses at 440 nm for sample excitation and was chopped at 500 Hz. The other generated an output at 1250 nm. The response of the sample to the 440 nm pump pulse was probed with a white light continuum generated by focusing into a sapphire plate either the 800 nm pulses (400–800 nm spectral window) or the 1250 nm pulses (500–1400 nm window). Pump and probe pulses were focused onto the sample flowing through a 1 mm fused silica flow cell, the pump being attenuated to 0.25 mW. The transmitted probe was detected by a home built polychromator and a CCD. The signals were detected at 1 kHz (pump-on and pump-off and a separate probe reference) and were processed to yield transient optical density difference spectra as described in Supplementary Note 5. The difference spectra were analyzed using global analysis software with a sequential model of bright FC state to dark state to metastable product, as described in Supplementary Note 6.

## Femtosecond stimulated Raman spectroscopy

The FSRS measurements were made using the same 800 nm source as for TA, which additionally pumped a second harmonic bandwidth compressor combined with a picosecond OPA. This produced a visible wavelength tuneable output (picosecond Raman pump) of tens to hundreds of milliwatts with a spectral width of 10 cm$^{-1}$. This source was chopped at 250 Hz and overlapped with the 440 nm pump (chopped at 500 Hz) and the white light continuum in the 1 mm pathlength flow cell. The sample optical density was adjusted to 0.7 at the excitation wavelength. The stimulated Raman signal was detected with a 10 cm$^{-1}$ resolution spectrometer and CCD detector to yield a spectral resolution of 20 cm$^{-1}$. Spectra were measured as a function of the pump 440 nm pulse delay time at 1 kHz. The four spectra which result from the double modulation scheme were combined to yield the FSRS spectra, as detailed in Supplementary Note 7. The picosecond OPA pulses were tuned to 560 nm, 650 nm and 700 nm to obtain the distinct resonance conditions described above.

## Data availability

The relevant experimental data that support the findings of this study have been deposited in the Source Data file. Additional data generated during this study are available in numerical format from the corresponding author upon request. Source data are provided with this paper.

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

## Acknowledgements

Financial support provided by The Netherlands Ministry of Education, Culture and Science (Gravity Program 024.001.035 to WRB, BLF) and the EPSRC (grants EP/R042357/1, EP/J009148/1 to SRM) is acknowledged.

## Author contributions

B.L.R., W.R.B., and S.R.M. conceived and designed the project. P.R. made all the sample preparation, measurements, data collection, analysis and Raman calculations. P.R. and S.R.M. drafted the paper. All contributors discussed, edited and refined the paper.

## Competing interests

The authors declare no competing interests.

## Additional information

**Peer review information** : *Nature Communications* thanks the anonymous reviewer(s) for their contribution to the peer review of this work. Peer reviewer reports are available.

