## [Peer Review File · Nature Communications]

Ultrafast Motion in a Third Generation Photomolecular MotorREVIEWER COMMENTS

Reviewer #1 (Remarks to the Author):

Review of Roy et al.

This manuscript reports a comprehensive experimental characterization of a third generation photomolecular motor. It uses a range of complementary experimental techniques together with computations at the DFT level.

Detailed comments:

1. In figure 1: what do the fat lines on the structures refer to? They are not really explained in the figure caption. Is this a schematic or does this figure - in particular panel b - report actual structures that were observed in the present work?
2. At the end of the Introduction it is stated "Here we use a combination of...". It is not mentioned that computations are used all along the way and without the computations most likely very little interpretation would be possible.
3. How precisely were the structures of the PSS and metastable forms associated with the experimentally observed spectra? Did the authors compute excitation spectra for the different structures? What are the barriers for rotation (as indicated in Figure 1b)?
4. On p. 9 the authors mention investigation of the "structural evolution". Which of the experimental techniques they used is directly sensitive to "structure as a function of time"? Comparison of the measured and computed Raman spectra is certainly not sufficiently robust for this, see SI Figure 17. In this figure it would be appropriate to report only the line positions from the computations also because the computed intensities are obviously not very useful.
5. pp. 12/13 states that "Based on ground state DFT..." Where precisely is this assignment reported? Is this based on frequency alone or also based on the nature of the normal mode? What are the experimental and computed frequencies that are used for identification. How reliable are the DFT calculations for assignment of modes given that the molecule is large and bands start to develop in the 1000 to 1700 cm⁻¹ region?
6. The text on p. 6 states that there is a 4.1±0.5 ps component from the TA whereas on p. 13 it is said that the 4.2 ps from TA was fixed. Which value was used in the global analysis and where does that 10 ns time come from?
7. Why are the relaxation times from FSRS close but not identical to those from TA (0.18 and 6 ps vs. 0.2 and 4.1/4.2 ps)?

8. The conclusion on p. 18 states that "FSRS measurements provide details on the structural evolution..". It remains unclear what the authors mean when referring to "structural evolution" throughout the manuscript. The experimental techniques they apply can be used to follow the "structural dynamics" of the compound under consideration, but do not directly report on the actual structure of the molecule. The whole point of this study appears to be the characterization of the underlying motions involved in the structural dynamics in order to make suitable chemical modifications to the compounds of interest so that they can be used as photochemically driven molecular motors. However, it remains unclear how the important intermediate structures are assigned from the experiments and whether the way this has been done (probably by comparing with DFT calculations) is meaningful.

In summary, although a comprehensive experimental characterization of the compound of interest was carried out, it remains unclear how the relevant molecular structures were assigned for the important steps along the photochemically driven pathway. The work as such is certainly a valid experimental study but lacks the compelling and profound molecular-level understanding required for a high-impact journal.

Further notes:

112: remains -> remain

203: filledexcited - this entire sentence is not very understandable.

Reviewer #2 (Remarks to the Author):

This paper reports an ultrafast electronic and vibrational spectroscopic study of a third-generation molecular motor (3GM) that is capable of photochemically driven unidirectional rotations.

A combination of femtosecond time-resolved emission and complementary absorption measurements was utilized to examine the initial electronic state dynamics. These measurements were conducted carefully over a wide spectral region, and the obtained data were adequately analyzed, fully characterizing the initial dynamics from the photogenerated bright state to the less emissive dark state with rich enough spectral information. Although the obtained picture is the one that is typically seen in various isomerizing molecules, the present data are indispensable to obtain a clue to understand the mechanism (driving force) and time scale of the initial motion of the molecular machine.

The authors subsequently examined ultrafast structural evolution of photoexcited 3GM by FSRS, which is expected to provide a structural insight. The FSRS measurements were extensively performed with two different Raman pump wavelengths. The contribution of the ground state molecule was reasonably subtracted from the raw data by the bleach filling method. Unfortunately, however, the resultant FSRS spectra in the C=C stretch region observed in the early time region exhibit a highly asymmetric band shape. Moreover, it seems that the band shape can be even a dispersive shape, depending on how the broad baseline is subtracted. Because of this ambiguity of the spectral analysis in FSRS, it is difficult to vibrationally characterize the transient state, particularly when such a complicated

band shape is observed. Actually, the authors decomposed the asymmetric band into two gaussian-shaped components, and discussed the localization of electronic excitation on one axle moiety. Although this argument constitutes one of key structural insights, the underlying spectral analysis is not reliable enough. It is well known that the asymmetric or dispersive band shape is often observed when the Raman pump is resonant with the stimulated emission transition, and it is exactly the case here. The asymmetric band shape can be merely a result of such spectral distortion, rather than the sum of the two spectral contributions. In fact, the direction of the asymmetric shape is inverted before and after 1 ps, when the origin of the stimulated emission changes from the bright state to the dark state. Thus, the apparent downshift from 1565 cm⁻¹ (0.2 ps) to 1539 cm⁻¹ (2 ps) is affected by the spectral distortion that reflects the resonant stimulated emission transition. Considering the importance of this structural argument in the present study, it is desirable to perform FSRS measurements using the Raman pump at around 1000 nm, where the stimulated emission is negligible but the excited state absorption is still observed strongly, and to reconsider a structural scenario through straightforward analysis of undistorted spectra.

As mentioned above, I recommend substantial revision of this manuscript, including the reexamination of the experimental conditions, before considering possible publication in Nature Communications. Additional comments are also listed below, which might help the authors improve the manuscript.

1) In time-resolved fluorescence up-conversion measurement, the authors observed oscillatory features due to coherent vibrations, and the signal showed in-phase oscillation at all wavelengths. In this type of measurement, the time origin can be shifted, depending on the wavelength, due to the group delay dispersion of the optics, and hence it needs to be compensated for in evaluating the phase relationship. How was the time origin determined at each wavelength?

2) In Figure 2c, is the horizontal axis label, especially “1000”, correct? It seems that the peak wavelength of the transient band observed in the near-infrared region is shifted when compared to the spectral data shown in the inset.

Reviewer #3 (Remarks to the Author):

Summary and general position of the ms. The present manuscript by the teams of B. Feringa and S. Meech reports on an extensive study of a so-called 3rd generation photo-molecular motor, designed in Ben Feringa's team as an evolution of the well-known directional chiral molecular rotors completing a full rotation after a sequential absorption of 2 photons. The present 3GM is designed with two axles and two rotors with the idea that the alternating rotation of both sides would allow for a linear “crawling” motion of the motor. While the synthesis, detailed structural and photo-chemical characterization was already disclosed (refs. 13&14), the present ms reports detailed results of ultrafast spectroscopy including UV/VIS transient absorption (TAS), femtosecond fluorescence up-conversion (TRUC) and stim. Raman spectroscopy (FSRS).

The aim being to find synthetic routes to enhance the photo-isomerisation quantum yield (PIQY), the main conclusions drawn from an impressive set of data concern the relation between the electronic (charge transfer) and vibrational properties of the so-called “dark” excited state intermediate and the PIQY. To this end, the target molecule is investigated in two solvents, acetonitrile (ACN) and cyclohexane (CHX), which feature different polarities and viscosities (ACN: 0.334 mPa s; CHX: 0.93 mPa·s at 22 °C).

General evaluation. The femtosecond techniques as well as the data analysis used in S.

Meech's team are first class, and the presented data are very clean and meaningful. Unfortunately, the photo-chemical scenario is more complicated than assumed. And hence the conclusions drawn from the observation are not as solid as presented.

First, the long-time TAS difference spectra alone, do not allow to determine the PIQY, even not in a relative fashion, like when comparing the same molecule in two solvents. As explained below, this has to be paralleled with NMR data of illuminated cis-trans mixtures. Second, the effect of changing ACN into CHX is discussed solely on the basis of the solvent polarity, which indeed decreases by a factor of ≈ 10 . But the viscosity also decreases significantly (3x), an effect, which is not discussed at all. The conclusions then focus on the dark state and its charge-transfer character. In my opinion, a more extensive study of a larger set of solvents is needed to substantiate the hypothesis along which the photo-induced CT of the dark state would be the key parameter to be addressed/scrutinised in future design efforts.

In conclusion, I recommend publication after major revisions (new experiments).

A full report including suggestions for a resubmission is uploaded as a pdf file.

Referee report on manuscript NCOMMS-22-20418

« Ultrafast Motion in Third Generation Photomolecular Motors » by P. Roy et al.

Summary and general position of the ms. The present manuscript by the teams of B. Feringa and S. Meech reports on an extensive study of a so-called 3rd generation photo-molecular motor, designed in Ben Feringa's team as an evolution of the well-known directional chiral molecular rotors completing a full rotation after a sequential absorption of 2 photons. The present 3GM is designed with two axles and two rotors with the idea that the alternating rotation of both sides would allow for a linear "crawling" motion of the motor. While the synthesis, detailed structural and photo-chemical characterization was already disclosed (refs. 13&14), the present ms reports detailed results of ultrafast spectroscopy including UV/VIS transient absorption (TAS), femtosecond fluorescence up-conversion (TRUC) and stim. Raman spectroscopy (FSRS).

The aim being to find synthetic routes to enhance the photo-isomerisation quantum yield (PIQY), the main conclusions drawn from an impressive set of data concern the relation between the electronic (charge transfer) and vibrational properties of the so-called "dark" excited state intermediate and the PIQY. To this end, the target molecule is investigated in two solvents, acetonitrile (ACN) and cyclohexane (CHX), which feature different polarities and viscosities (ACN: 0.334 mPa s; CHX: 0.93 mPa·s at 22 °C).

General evaluation. The femtosecond techniques as well as the data analysis used in S. Meech's team are first class, and the presented data are very clean and meaningful. Unfortunately, the photo-chemical scenario is more complicated than assumed. And hence the conclusions drawn from the observation are not as solid as presented.

First, the long-time TAS difference spectra alone, do not allow to determine the PIQY, even not in a relative fashion, like when comparing the same molecule in two solvents. As explained below, this has to be paralleled with NMR data of illuminated cis-trans mixtures.

Second, the effect of changing ACN into CHX is discussed solely on the basis of the solvent polarity, which indeed decreases by a factor of ≈ 10 . But the viscosity also decreases significantly (3x), an effect, which is not discussed at all. The conclusions then focus on the dark state and its charge-transfer character. In my opinion, a more extensive study of a larger set of solvents is needed to substantiate the hypothesis along which the photo-induced CT of the dark state would be the key parameter to be addressed/scrutinized in future design efforts.

Main problems with assumptions, interpretation of data and in conclusions.

- a) The main assumption used in the temporal analysis of the data and their casting into evolution-associated difference spectra is to say that the photo-reaction is purely sequential with only three states: bright ES \rightarrow dark ES \rightarrow cis/trans GS mixture (metastable state).

How can this be justified ?

One may imagine that internal conversion bright ES \rightarrow metastable cis/trans mixture occurs in parallel to bright ES \rightarrow dark ES \rightarrow .

And, why exclude a scenario such as: bright ES \rightarrow dark ES \rightarrow vibr. "hot" cis/trans GS mixture \rightarrow vibr. rel. cis/trans GS mixture. The formation of vibr. hot GS species either directly from the bright ES or the dark ES are plausible since the lifetimes of these

states are shorter than typical vibration relaxation times (5-10 ps). This would then give a very different interpretation of the short-time FSRS difference spectra. A comment from the authors would be appreciated.

- b) A relative comparison of the PIQY from the amplitudes of ΔA data in different solvents is a very shaky approach. It could work only if the shape of the ss-A and the ΔA spectra are strictly superposable since this would imply that the photo-product (PP) spectrum is also solvent-independent.

In the present case, these two conditions are not given. Surprisingly, in ACN, the bleach amplitude is almost zero, while in cHex it is not (fig. 4b). Clearly indicating solvent-dependent shifts of PP spectra. The reduced ΔA amplitude at 530 nm can simply be due to a higher spectral overlap (smaller red-shift, less broadening) of the ss and PP absorption spectra in ACN.

Since solvent dependence of the PIQY is one of the main conclusions of the paper, the author should use the appropriate methods to demonstrate it. This requires the determination of the absorbance spectrum of the metastable state from a photo-stationary mixture. A quantitative comparison of NMR spectra – identifying the isomer content – and the absorption spectra of the mixture (best done for different photo-stationary cis/trans compositions) will then allow to determine the PP absorption spectrum in detail.

The ΔA data can then be analysed/decomposed so as to yield the PIQY.

- c) the effect of changing ACN into CHX is discussed solely on the basis of the solvent polarity, and effect of viscosity is neglected. Why? Can the authors motivate their assumption?

In the present form, the conclusion stating that the photo-induced CT of the dark state would be the key parameter controlling PIQY is not sufficiently justified. A weaker, less conclusive formulation would be more appropriate. And it is highly recommended to substantiate this claim by a study of a larger set of solvents in an up-coming paper.

- d) Raman spectra: Can you justify why the 1553cm^{-1} mode is so much more intense than all other modes?

- e) The blue-shifted excited state Raman frequencies are assigned to C=C vibrations of the axle, and interpreted as an increased local bond order. Isn't that completely counter-intuitive with the fact that rotation occurs around these bonds? I had expected the opposite effect since optical excitation is of pi-pi* character, i.e. it weakens the bond order of the axles thus allowing for photo-induced rotation around these axles. Please comment. An alternative interpretation may be that rotor and stator become electronically decoupled in the excited state, and so do the normal modes of vibrations. That leads to two distinct modes and a blue-shift due a smaller reduced mass for either mode.

- f) What is a "non-Condon" effect? Do you mean "non-Born Oppenheimer" or an effect "beyond the Franck-Condon approximation"? Please avoid jargon, but use common text book terminology.

- g) Figures 18 and 19 are really difficult to read. The symmetric/anti-symmetric character of the two HF modes in fig. 18 is not obvious (arrow heads too small). Atoms superpose, but the images do not have perspective. Use different color to differentiate the rotor and stator atoms.
- h) Proof-reading by a native speaker will allow eliminating residual typos. Some sentences are contracted, and link words like “that is” or “which is” are missing. Ex.

In conclusion. The manuscript NCOMMS-22-20418 contains a number of new and highly interesting results on the ultrafast spectroscopy of this 3rd generation motor. The complementary information gained by the electronic and vibrational (not “structural”, wrong phrasing in legend of fig. 3) spectroscopies warrant to be showcased in Nature Comm.

However, not in the present form. I strongly recommend to work along the above suggestions (solvent dependent PIQY) and weaken the conclusions if needed.

We are grateful to all three reviewers for their positive assessments and their helpful comments. We have endeavoured to fully address all of them, including by the addition of some new experiments and analysis (hence the delay in resubmission). We have successfully separated polarity and viscosity effects and confirmed the relation between long time transient absorption and metastable state yield. We also conducted the best experiment we could on red-shifted FSRS, substantially refined the discussion on the lineshape analysis and highlighted our underlying assumptions.

In the following we have exactly reproduced the reviewer's comments, editing them only to add a numbering scheme to allow cross referencing. The comments are normal typeface (marked by blue color), our replies are italic and new text in red (as it also appears in the revised manuscript) and the text it replaces is shown in plain black text.

Reviewer #1 (Remarks to the Author):

Review of Roy et al.

This manuscript reports a comprehensive experimental characterization of a third generation photomolecular motor. It uses a range of complementary experimental techniques together with computations at the DFT level.

Detailed comments:

1.1. In figure 1: what do the fat lines on the structures refer to? They are not really explained in the figure caption. Is this a schematic or does this figure - in particular panel b - report actual structures that were observed in the present work?

The lines are standard projections in organic chemistry but for a broader audience we agree we should have noted that they indicate atoms above or below the plane. Structures have been determined elsewhere by crystallography, NMR and calculation (now cited).

We add to the legend of Figure 1

; thicker lines indicate atoms above the plane of the page, light lines below

And

(structures from Kistemaker et al. [14])

1.2. At the end of the Introduction it is stated "Here we use a combination of...". It is not mentioned that computations are used all along the way and without the computations most likely very little interpretation would be possible.

Agreed, on P5 we now highlight our reliance on DFT in the final sentence of the introduction

with analysis of the latter supported by density functional theory calculations; details of experiments and calculations...

1.3. How precisely were the structures of the PSS and metastable forms associated with the experimentally observed spectra? Did the authors compute excitation spectra for the different structures? What are the barriers for rotation (as indicated in Figure 1b)?

The ground state Raman spectra were measured and compared with DFT calculations of the two possible isomers (Figure 3a,b and associated discussion) leading to the assignments. The metastable state's red-shifted electronic spectrum was inferred from low temperature measurements reported previously and shown and discussed in Supplementary Figure 7. The barriers for the ground state rotation were not determined here – they were reported earlier [14] for other 3GMs as 50 – 70 kJmol⁻¹ and we expect the present derivative to be in the same range, consistent with the second timescale relaxation of the metastable form (see below). It is an important note that the focus here is on the ultrafast photocycle rather than the slow thermally driven helix inversion; extensive temperature dependent measurements of ground state relaxation are required for further information.

1.4. On p. 9 the authors mention investigation of the "structural evolution". Which of the experimental techniques they used is directly sensitive to "structure as a function of time"? Comparison of the measured and computed Raman spectra is certainly not sufficiently robust for this, see SI Figure 17. In this figure it would be appropriate to report only the line positions from the computations also because the computed intensities are obviously not very useful.

Vibrational spectroscopy is the structural method of choice here. Interpreting the reviewer's comments somewhat, we agree that the level of structural detail available from Raman spectroscopy is low compared to crystallography for example. However, we have had significant success correlating Raman with structure (for a recent example see <https://www.nature.com/articles/s41565-019-0401-6>). More significantly, crystallography is not a dynamic method (apart from at a few FEL facilities) and the structural dynamics of importance in molecular machines will certainly be suppressed in crystals. Probably NMR can also provide better structural information than Raman spectroscopy, but again it does not have the time resolution needed to study excited state reactions. Consequently, for experimental studies of fast structural evolution in chemistry and biology one invariably relies on ultrafast time resolved vibrational spectroscopies (ideally accompanied by quantum chemical calculations, although these remain very difficult for excited states of large molecules). Thus, vibrational spectroscopy is the only structural tool for fast measurements in fluids. For a broad audience we should probably have indicated the limitations of the choice of Raman as a structural tool. The concluding sentence of the introduction, already *amended* (point 1.2 above) was:

We characterise its light driven reaction through ultrafast fluorescence, transient absorption (TA) and femtosecond stimulated Raman spectroscopy (FSRS) *with analysis of the latter supported by density functional theory calculations*; details of experiments and calculations are provided in Supporting Information.

Has now been further replaced by

Ultrafast fluorescence and transient absorption characterise population relaxation and coherent vibrational dynamics in 3GMph. Femtosecond stimulated Raman spectroscopy (FSRS) adds structural dynamics data at the level of vibrational spectroscopy, with its analysis being supported by density functional theory calculations. Details of experiments and calculations are provided in Supplementary Information.

Concerning the comparison of DFT and experimental Raman, we agree that there is no direct way of correlating the intensity of all modes, since the FSRS provides pre-resonant Raman spectra while DFT calculation provides non-resonant Raman spectra. However the fundamental nature of the Raman modes remain unchanged. Therefore, we would prefer to keep the calculated Raman spectra instead of showing only the lines. In fact amplitudes as well as frequencies for the main modes considered

>1000 cm^{-1} are reasonably well reproduced, and this has been our experience in the past e.g. <https://pubs.acs.org/doi/full/10.1021/jp404925m>.

1.5a. pp. 12/13 states that "Based on ground state DFT..." Where precisely is this assignment reported? Is this based on frequency alone or also based on the nature of the normal mode? What are the experimental and computed frequencies that are used for identification.

The mode assignments were based on the normal mode picture, which shows the CC stretch as the dominant nuclear coordinate for these frequencies; this was mentioned in discussion of the ground state Raman. We now make the connection clearer in the relevant section by adding on p13

(the ground state modes at 1554 and 1540 cm^{-1} are shown in Figure S18)

1.5b How reliable are the DFT calculations for assignment of modes given that the molecule is large and bands start to develop in the 1000 to 1700 cm^{-1} region?

On this broader question, DFT is a robust widely used method for calculating nuclear structure, energy and vibrational and NMR spectra in large organic molecules. It has been very well tested over the past 20 years, and for the ground electronic state the high frequency range (>1000 cm^{-1}) are accurately recovered and assigned (as indicated by our own isotope studies, Roy et al, J. Phys. Chem. Lett. 2021, 12, 13, 3367 and many others). Amplitudes are usually well reproduced at least for localised modes. In our experience for this and much larger molecules ground state vibrational spectra are robust and reliable.

1.6. The text on p. 6 states that there is a 4.1+/-0.5 ps component from the TA whereas on p. 13 it is said that the 4.2 ps from TA was fixed. Which value was used in the global analysis and where does that 10 ns time come from?

Apologies – a typo (it should be 4.1), now corrected (p13). The query regarding '10 ns' is addressed in response to point 1.7

1.7. Why are the relaxation times from FSRS close but not identical to those from TA (0.18 and 6 ps vs. 0.2 and 4.1/4.2 ps)?

This is a signal to noise issue, which arises from the weaker FSRS signal and the long time it takes to measure the spectra at each pump-probe time delay (which means we are limited in how many pump-probe delay points we can measure). For this reason TA time constants are preferred. We now clarify this point by replacing the text from p13

To resolve Raman spectra associated with the distinct states, we performed global analysis with fixed time constants taken from TA data (0.2 ps, 4.1 ps and 10 ns). This recovers evolution...

With

To resolve Raman spectra of the distinct states we performed global analysis with fixed time constants taken from TA data (0.2 ps, 4.1 ps and 10 ns). **The 4.1 ps time constant from TA global analysis is assumed to be more accurate than the 6 ps from fitting the FSRS data, as the TA data have better signal-to-noise and were measured at a larger number of time delays in the picosecond time range. An arbitrary long time of 10 ns is selected to capture the final long-lived metastable state contribution. This global analysis recovers**

1.8. The conclusion on p. 18 states that "FSRS measurements provide details on the structural evolution.". It remains unclear what the authors mean when referring to "structural evolution" throughout the manuscript. The experimental techniques they apply can be used to follow the "structural dynamics" of the compound under consideration, but do not directly report on the actual structure of the molecule. The whole point of this study appears to be the characterization of the underlying motions involved in the structural dynamics in order to make suitable chemical modifications to the compounds of interest so that they can be used as photochemically driven molecular motors. However, it remains unclear how the important intermediate structures are assigned from the experiments and whether the way this has been done (probably by comparing with DFT calculations) is meaningful.

As mentioned above, vibrational spectroscopy is usually regarded as a structural method, and is the only one available for light driven reactions in the most important media for chemistry (liquids or cells). We are however happy to either highlight the admitted limitations of vibrational spectroscopy or to refer to 'structural dynamics' rather than 'structural evolution' as appropriate at each occurrence if that terminology is preferred.

In the abstract we replace structural evolution with structural dynamics; 'structural evolution' seems appropriate on p5; on p9 we again change to structural dynamics and extend our description with 'structural dynamics, we apply resonant FSRS to recover transient vibrational spectra, which can provide insight into intermediate state structures' to highlight the level of structural information attainable; on p18 we prefer to replace 'structural evolution in the 3GM photocycle' with structural evolution in the 3GM photocycle (at the level of vibrational spectroscopy) to highlight the aforementioned limitations of Raman.

In summary, although a comprehensive experimental characterization of the compound of interest was carried out, it remains unclear how the relevant molecular structures were assigned for the important steps along the photochemically driven pathway. The work as such is certainly a valid experimental study but lacks the compelling and profound molecular-level understanding required for a high-impact journal.

*Here we can only reiterate the point that the structural method we rely on - vibrational spectroscopy - is the **only** structure tool available for excited state reactions. It is for sure imperfect, but when coupled with high level excited state quantum calculations it has repeatedly been shown to yield meaningful structures. Such calculations are beginning to appear for motors, and the novel high-quality data we have obtained (including temporal evolution in motor excited state vibrational structure never previously resolved) will stand as a challenge to them. In this way we can make progress in controlling molecular motors. We believe this is important progress and warrants publication in a high-impact journal.*

Further notes:

112: remains -> remain

Thank you – now remain

203: filled excited - this entire sentence is not very understandable.

Agreed – but this bleach filling and data analysis section is rewritten in response to other reviewer comments, so is addressed below (see 2.2).

Reviewer #2 (Remarks to the Author):

2.1 This paper reports an ultrafast electronic and vibrational spectroscopic study of a third-generation molecular motor (3GM) that is capable of photochemically driven unidirectional rotations. A combination of femtosecond time-resolved emission and complementary absorption measurements was utilized to examine the initial electronic state dynamics. These measurements were conducted carefully over a wide spectral region, and the obtained data were adequately analyzed, fully characterizing the initial dynamics from the photogenerated bright state to the less emissive dark state with rich enough spectral information. Although the obtained picture is the one that is typically seen in various isomerizing molecules, the present data are indispensable to obtain a clue to understand the mechanism (driving force) and time scale of the initial motion of the molecular machine.

A fair summary – thank you.

2.2a The authors subsequently examined ultrafast structural evolution of photoexcited 3GM by FSRS, which is expected to provide a structural insight. The FSRS measurements were extensively performed with two different Raman pump wavelengths. The contribution of the ground state molecule was reasonably subtracted from the raw data by the bleach filling method. Unfortunately, however, the resultant FSRS spectra in the C=C stretch region observed in the early time region exhibit a highly asymmetric band shape. Moreover, it seems that the band shape can be even a dispersive shape, depending on how the broad baseline is subtracted. Because of this ambiguity of the spectral analysis in FSRS, it is difficult to vibrationally characterize the transient state, particularly when such a complicated band shape is observed. Actually, the authors decomposed the asymmetric band into two gaussian-shaped components, and discussed the localization of electronic excitation on one axle moiety.

The reviewer is absolutely correct in saying that the correction method to the complex FSRS lineshapes we have chosen (bleach filling) is not the only possibility. We were aware of other contributions and presented evidence in supplementary information that bleach filling was the most plausible assignment. We chose supplementary pages for this as we did not want to weigh down the main text of a ‘broad interest’ paper with this detailed discussion of lineshape analysis. However we agree with the reviewer that we have erred in then discussing in the main paper only that possibility. We have now added more balanced discussion of our choices in the lineshape analysis to the main text and as a result changed the discussion to soften our assignment (as also suggested by reviewer 3). We believe our raw data (especially Raman pump at 650 nm) show that the lineshape we observed is real (there is in fact no hint of a dispersive lineshape with Raman pump at 650 and 700 nm, although the lineshape at early times is the same as that at 560 nm– see supplementary data). The following changes are also relevant to 2.2b, below.

The text on P11 read

In addition to the expected positive stimulated Raman signal enhanced by the S_1 to S_n transient absorption, contributions from S_1 to S_0 stimulated emission (variously described as Raman induced nonlinear emission and inverse Raman) and S_0 to S_1 pre-resonance Raman all lead to negatively signed contributions at or near the wavenumber of GSR modes

Now reads

In addition to the expected positive stimulated Raman signal enhanced by the S_1 to S_n transient absorption, contributions from S_1 to S_0 stimulated emission (variously described as Raman induced nonlinear emission, RINE, or inverse Raman) and S_0 to S_1 pre-resonance Raman all lead to negatively signed contributions at or near the wavenumber of GSR modes. These two separate contributions lead to distinct signals in addition to the positive S_1 Raman spectrum sought, one (RINE) characterized by a differential lineshape and one (S_0 - S_1 bleach feature) by a negative contribution from the ground state Raman. Mathies and...

Later the text continued

with the negative feature largely suppressed; thus, this spectrum can be assigned to the bright state. In addition, we observed that the GSR signal was also reduced four-fold, as expected when moving the Raman-pump further to the red in the pre-resonance regime. The pump wavelength dependence of ESR₅₆₀ suggests the dominant contribution to the negative signal arises from GSR. Such a feature can be corrected by quantitative bleach filling, *i.e.* adding controlled amounts of the measured GSR to the ESR₅₆₀ data.[39] Initially we fill the 760 ps data, where only product and bleach contribute, to recover a smooth product Raman spectrum (Supplementary Figure 12a). We then add to all ESR₅₆₀ data a fraction of the GSR determined by the bleach population recovered from TA. (Figure 2d and Supplementary Figure 12b). Thus, the bleach-filled excited state Raman data in Figure 3c and 3d are obtained; equivalent data were recovered from ESR₆₅₀ (Supplementary Figure 13, 15)..

Which is now rewritten and extended:

with the negative feature largely suppressed, although its contribution is still resolvable in the long-time data as a weak negative feature (compare Supplementary Figures 9 and 10). An even better experiment would be the use of a Raman pump beyond 1 micron (where the broad SE is negligible, Figure 2a,c). However, current detectors provide insufficient signal-to-noise. Hence, we have been unable to measure with a Raman pump wavelength >700 nm, and those data did not show a RINE feature at early time (Supplementary Figure 21), which was essentially identical to the 650 nm data. In addition, we observed that the GSR signal was also reduced four-fold at 650 nm compared to 560 nm, as expected when moving the Raman-pump further to the red in the pre-resonance Raman regime. Two features of these data suggest that the dominant contribution to the negative amplitude is the bleached GSR signal. First, the negative feature persists at all delay times, whereas the RINE features usually decay on the sub-picosecond timescale, faster than the population decay.[36,39] Secondly, even though the negative feature is well resolved at long times (especially with the 560 nm Raman pump) there no sign of the positive lobe invariably associated with the differential lineshape from RINE. If we assume the dominant contribution is indeed bleached GSR, the negative contribution can be corrected by quantitative bleach filling, *i.e.* adding controlled amounts of the measured GSR to the ESR₅₆₀ data, as has been described elsewhere.[39-42] Initially we fill the 760 ps data, where only product and bleach contribute, to recover a smooth product Raman spectrum (Supplementary Figure 12a). We then add to all ESR₅₆₀ data a fraction of the GSR determined by the bleach population amplitude recovered from TA (Figure 2d and Supplementary Figure 12b). Thus, the bleach-filled excited state Raman data in Figure 3c and 3d are obtained; equivalent data were recovered from ESR₆₅₀ (Supplementary Figure 13, 15). This successful quantitative application of the bleach filling model points to the GSR bleach being the main factor distorting the lineshape of the S_1 Raman spectrum, which has then allowed us to isolate the desired spectrum (Figure 3c,d). The dominance of the GSR contribution at 1555 cm^{-1} is consistent with the observation of an intense GSR signal at this frequency. However, although we do not require any additional RINE contributions to fit the present lineshape we cannot rule out the possibility of a contribution at early times, as discussed below.

We hope this expanded text represents a more balanced presentation of the possible contributions to the lineshape and the assumption we made in our analysis. Next we turn to the effect this ambiguity might have on the discussion of our results. We recognize this by adding on p13

Although two modes are predicted in this C=C stretch region we cannot completely rule out that the distinctive lineshape observed has contributions from uncorrected RINE features (although the facts that the amplitude of bleach filling required was quantitatively predicted by the population from TA, and that the 650 and 700 nm raw FSRS data show no hint of a differential lineshape at early times, yet the spectrum recovered at < 1 ps is identical to that at 560 nm, see Supplementary Figures 21, both point to a dominant bleach filling mechanism). In the case that RINE does contribute to the lineshape it would simply be one broad and the exact frequencies of the two modes less certain. What is clear...

To discussion on p14 we simply add

see above and

2.2b Although this argument constitutes one of key structural insights, the underlying spectral analysis is not reliable enough. It is well known that the asymmetric or dispersive band shape is often observed when the Raman pump is resonant with the stimulated emission transition, and it is exactly the case here. The asymmetric band shape can be merely a result of such spectral distortion, rather than the sum of the two spectral contributions. In fact, the direction of the asymmetric shape is inverted before and after 1 ps, when the origin of the stimulated emission changes from the bright state to the dark state. Thus, the apparent downshift from 1565 cm⁻¹ (0.2 ps) to 1539 cm⁻¹ (2 ps) is affected by the spectral distortion that reflects the resonant stimulated emission transition. Considering the importance of this structural argument in the present study, it is desirable to perform FSRS measurements using the Raman pump at around 1000 nm, where the stimulated emission is negligible but the excited state absorption is still observed strongly, and to reconsider a structural scenario through straightforward analysis of undistorted spectra.

We have addressed the complexity of the lineshape above, Indeed we agree that would be a perfect experiment (and indicated that – see response to 2.2a). However, it is not currently possible. The most serious limitation is in signal-to-noise. The detector and the continuum intensity in the NIR are adequate for TA but not for FSRS, as the background subtraction protocol (supplementary data Figs 9 and 10) has very high demands on both signal-to-noise and continuum stability. In particular the monochromator/CCD sensitivity drops sharply in the NIR. Thus, to shift to the NIR we would need to replace our detectors and optimise continuum (we tried YAG as a replacement for sapphire, but it did not help significantly). In addition, it would be necessary to realign the internal optics of the Raman OPA and replace all of the visible coated reflective optics in the beam path. Thus, progress in this desirable direction is subject to the uncertainty of future research funding, and the measurement requested is not currently possible.

We have tried going further to the red, but the limit in terms of useful signal to noise the limit was 700 nm Raman pump (Stokes FSRS signals to > 790 nm). The text (see 2.2a) is amended as follows:

An even better experiment would be the use of a Raman pump beyond 1 micron (where the broad SE is negligible, Figure 2a) but current detectors provide insufficient signal-to-noise. Hence, we have been unable to measure with a Raman pump wavelength >700 nm, and those data did not show a RINE feature at early time (Supplementary Figure 21).

And the data is shown in Supplementary Figure 21 and reproduced below

Supplementary Figure 21: a) Raw excited state Raman data (ESR, Raman pump at 700 nm) of 3GMph in ACN at different time delays with the corresponding baselines (blue) are shown. b) The baseline corrected raw ESR spectra are shown. c) The bleach filled ESR spectra at different time delays are shown. Asterisk (*) shows the region of solvent artifact. d) The corresponding bleach filling amount at different time delays are plotted (marked by red star). The bleach filling dynamics from FSRs follows the ground state bleach recovery dynamics from TA (black line).

As mentioned above, I recommend substantial revision of this manuscript, including the reexamination of the experimental conditions, before considering possible publication in Nature Communications. Additional comments are also listed below, which might help the authors improve the manuscript.

2.3 In time-resolved fluorescence up-conversion measurement, the authors observed oscillatory features due to coherent vibrations, and the signal showed in-phase oscillation at all wavelengths. In this type of measurement, the time origin can be shifted, depending on the wavelength, due to the group delay dispersion of the optics, and hence it needs to be compensated for in evaluating the phase relationship. How was the time origin determined at each wavelength?

This is an important experimental point. We determine time zero by measuring the up-conversion of the (instantaneous) solvent Raman at a wavelength of 470 nm, near the emission. In earlier work

(the cited method paper) we measured the dispersion of different Raman frequencies and found it negligible (probably because we use reflective optics wherever possible, so only the sample cell contributes). We note explicitly in the Supplementary description, where our recompression of excitation and gate beams with dispersive mirrors is also described.

The IRF and time zero for the experiment was determined to be around 43 fs by recording the upconversion of Raman scattering from heptane at 470 nm. All reflective optics were used to minimise dispersion and pulse broadening.

2.4 In Figure 2c, is the horizontal axis label, especially “1000”, correct? It seems that the peak wavelength of the transient band observed in the near-infrared region is shifted when compared to the spectral data shown in the inset.

Thank you for spotting this error - ‘1000 nm’ in Figure 2c should have been ‘900 nm’. We have now corrected Figure 2c.

Reviewer #3 (Remarks to the Author):

These are copied in full from the longer pdf supplied

Summary and general position of the ms. The present manuscript by the teams of B. Feringa and S. Meech reports on an extensive study of a so-called 3rd generation photo-molecular motor, designed in Ben Feringa's team as an evolution of the well-known directional chiral molecular rotors completing a full rotation after a sequential absorption of 2 photons. The present 3GM is designed with two axles and two rotors with the idea that the alternating rotation of both sides would allow for a linear “crawling” motion of the motor. While the synthesis, detailed structural and photo-chemical characterization was already disclosed (refs. 13&14), the present ms reports detailed results of ultrafast spectroscopy including UV/VIS transient absorption (TAS), femtosecond fluorescence up-conversion (TRUC) and stim. Raman spectroscopy (FSRS). The aim being to find synthetic routes to enhance the photo-isomerisation quantum yield (PIQY), the main conclusions drawn from an impressive set of data concern the relation between the electronic (charge transfer) and vibrational properties of the so-called “dark” excited state intermediate and

the PIQY. To this end, the target molecule is investigated in two solvents, acetonitrile (ACN) and cyclohexane (CHX), which feature different polarities and viscosities (ACN: 0.334 mPa s; CHX: 0.93 mPa s at 22 °C).

General evaluation. The femtosecond techniques as well as the data analysis used in S. Meech's team are first class, and the presented data are very clean and meaningful. Unfortunately, the photo-chemical scenario is more complicated than assumed. And hence the conclusions drawn from the observation are not as solid as presented.

3.1 First, the long-time TAS difference spectra alone, do not allow to determine the PIQY, even not in a relative fashion, like when comparing the same molecule in two solvents. As explained below, this has to be paralleled with NMR data of illuminated cis-trans mixtures.

*Thank you – this is an important point which we need to clarify. Our assignment and the comparison is only justified if the ground state spectra of **both** stable and metastable states are solvent independent (in which case the difference is just the metastable state population, as we claim). Solvent independence was already established for the stable state (Supplementary Figure 1a). This point should have been better highlighted in the main text so we now write on p15 in place of*

Steady-state absorption and emission of 3GMph in CHX have maxima at 442 nm and 560 nm respectively

We write

The absorption spectra are essentially independent of solvent polarity, while the fluorescence in CHX is slightly blue shifted compared to ACN.

We now show this solvent independence is the case for the difference spectra (i.e. where there is a contribution corresponding to the absorption of metastable state) - which we have measured to be identical in a range of solvents. Only the amplitudes are solvent dependent, not the maxima or spectral profile. This result is only possible if the metastable state absorption spectrum is also solvent independent (not an unexpected result when the two structures are compared). These data are shown in Supplementary Figure 20c and reproduced below. The raw TA data of 3GMph in tetradecane and DMSO are also shown in Supplementary Figure 20a and 20b. To highlight this we add to the text on p16

The original text was

The metastable product absorption at 530 nm is doubled while the GSB is correspondingly increased, suggesting that the relative photoconversion yield in nonpolar vs polar solvent is on the order of 2:1.

Which have been modified to read

The metastable product absorption at 530 nm is doubled while the GSB is correspondingly increased. **Since only the metastable and stable states contribute to the difference**

spectrum at long times this solvent dependence can arise in only two ways; either the metastable state is solvent dependent and is shifted under the solvent independent (Supplementary Figure 1a) stable state spectrum or the yield of the metastable state from the dark state is solvent dependent. To resolve this, the long-time difference spectrum was measured in a range of solvents of different polarity (Supplementary Figure 20c). The shape and peak wavelength of the difference spectra are independent of solvent, while their amplitude is a strong function of solvent polarity, decreasing by a factor of 2 between CHX and ACN (Figure 4) and by a factor of 15 between CHX and dimethyl sulfoxide (DMSO), Supplementary Figure 20c).

Figure 20. Transient absorption spectra of 3GMph in a) tetradecane and b) DMSO are recorded at different pump-probe delays. c) TA spectra at 500 ps are plotted for comparison in solvents with different polarity and viscosity. The measurements were done under similar experimental conditions (sample concentration, pump power and overlap) in order to compare the relative yield of metastable product formation.

3.2 Second, the effect of changing ACN into CHX is discussed solely on the basis of the solvent polarity, which indeed decreases by a factor of ~ 10 . But the viscosity also decreases significantly (3x), an effect, which is not discussed at all. The conclusions then focus on the dark state and its charge-transfer character. In my opinion, a more extensive study of a larger set of solvents is needed to substantiate the hypothesis along which the photo-induced CT of the dark state would be the key parameter to be addressed/scrutinized in future design efforts.

Again this is an important point which we can now address with this wider range of solvents studied, which have a range of viscosities as well as polarities. The text in response to 3.1 Now continues:

The conversion from FC through dark state to the metastable form requires an isomerisation reaction, which may displace a significant solvent volume and so be sensitive to viscosity as well as polarity. The data in Supplementary Figure 20c show that polarity is the determining factor. For example, the viscosities of nonpolar CHX and tetradecane differ by a factor of 5 but when measured under identical conditions the metastable state yield is identical. In contrast, DMSO and tetradecane differ in viscosity by less than 10% but the metastable yield is reduced a factor of 15 in polar DMSO. This demonstrates that formation of the metastable state is suppressed in polar solvents, from which it is concluded that medium polarity is a potential method of modulating the yield of metastable state formation in 3GMph.

Main problems with assumptions, interpretation of data and in conclusions.

3.3 The main assumption used in the temporal analysis of the data and their casting into evolution-associated difference spectra is to say that the photo-reaction is purely sequential with only three states: bright ES \rightarrow dark ES \rightarrow cis/trans GS mixture (metastable state). How can this be justified? One may imagine that internal conversion bright ES \rightarrow metastable cis/trans mixture occurs in parallel to bright ES \rightarrow dark ES \rightarrow . And, why exclude a scenario such as: bright ES \rightarrow dark ES \rightarrow vibr. "hot" cis/trans GS mixture \rightarrow vibr. rel. cis/trans GS mixture. The formation of vibr. hot GS species either directly from the bright ES or the dark ES are plausible since the lifetimes of these states are shorter than typical vibration relaxation times (5-10 ps). This would then give a very different interpretation of the short-time FSRS difference spectra. A comment from the authors would be appreciated.

The guiding principle here is Occam's razor. Our raw data show explicitly that there is an intermediate (the FC state decays before the ground state recovers). We fit the data with one intermediate and obtain a good fit across the entire vis NIR bandwidth. Thus we have used the simplest model. Further, if we add extra terms no new states are resolved. The hot ground state model is interesting, but requires a new intermediate. None was found. This does not mean the metastable state is formed cold, but that the main part of vibrational relaxation in ACN/CHX is faster than the 4 / 12 ps, which is not unreasonable (it may also mean that the electronic spectrum is relatively insensitive to internal temperature after IVR). Other schemes may be drawn but they are all more complicated than the one used, and fit no better, so their use cannot be justified.

To indicate this we add (p8)

The same good fit of this simple model is found in all solvents studied and applies equally to the FSRS data (below).

3.4 A relative comparison of the PIQY from the amplitudes of ΔA data in different solvents is a very shaky approach. It could work only if the shape of the ss-A and the DA spectra are strictly superposable since this would imply that the photo-product (PP) spectrum is also

solvent-independent. In the present case, these two conditions are not given. Surprisingly, in ACN, the bleach amplitude is almost zero, while in cHex it is not (fig. 4b). Clearly indicating solvent-dependent shifts of PP spectra. The reduced DA amplitude at 530 nm can simply be due to a higher spectral overlap (smaller red-shift, less broadening) of the ss and PP absorption spectra in ACN.

This is the substance of the comments listed under 3.1 above and we refer to that answer. As correctly surmised here the ground state spectra are independent of solvent polarity, as we have now shown.

3.5 Since solvent dependence of the PIQY is one of the main conclusions of the paper, the author should use the appropriate methods to demonstrate it. This requires the determination of the absorbance spectrum of the metastable state from a photostationary mixture. A quantitative comparison of NMR spectra – identifying the isomer content – and the absorption spectra of the mixture (best done for different photo-stationary cis/trans compositions) will then allow to determine the PP absorption spectrum in detail.

The ΔA data can then be analysed/decomposed so as to yield the PIQY.

We and other have used NMR to assess conversion yields in 1 and 2GMs, but this is not possible for 3GMs as the metastable to stable state relaxation is too fast to be resolved at room temperature, so insufficient concentration is established. Nevertheless, we believe that the analysis in Supplementary Figure 20c has addressed all of these concerns. To go further we constructed a photochemical cell in a spectrometer to resolve the two electronic spectra for 3GMph by recording the absorbance of the photostationary state under irradiation by LEDs (as has been done for other 3GMs at reduced temperature, Nature Chemistry, 7, 890 (2015). This measurement (shown below for two different wavelengths) gave a negative result – the metastable to product state relaxation is too fast to build up a significant steady state population. The data are shown below in Figure A (but were not added in the Supplementary Information since it doesn't contain any positive signal).

Figure A. Absorption spectra of 3GM in cyclohexane in presence and absence of LED at a) 385 nm and b) 410 nm. The difference spectra (green) do not show any positive signature at 530 nm corresponding to metastable product. This indicates that the metastable product is short-lived.

In the original text we omitted to mention the metastable state was stabilised at low temperature, now corrected (p9)

at reduced temperature

3.6 the effect of changing ACN into CHX is discussed solely on the basis of the solvent polarity, and effect of viscosity is neglected. Why? Can the authors motivate their assumption? In the present form, the conclusion stating that the photo-induced CT of the dark state would be the key parameter controlling PIQY is not sufficiently justified. A weaker, less conclusive formulation would be more appropriate. And it is highly recommended to substantiate this claim by a study of a larger set of solvents in an upcoming paper.

The claim was substantiated in Supplementary Figure 20c

3.7 Raman spectra: Can you justify why the 1553 cm^{-1} mode is so much more intense than all other modes?

The factors (Albrecht A and B terms) which determine intensity of Raman scattering for a particular mode are well known but quite complicated, so it is difficult to select one feature to explain a relative amplitude of a particular mode (beyond the trivial statement that the slope of the polarizability near $Q = 0$ must be large). Although we might expect the S_1 displaced C=C mode to be resonance enhanced, this does not appear to be a resonance effect as DFT calculations (off resonance) and experiment agree on the large C=C amplitude. Under these circumstances it does not seem helpful to comment.

3.8 The blue-shifted excited state Raman frequencies are assigned to C=C vibrations of the axle, and interpreted as an increased local bond order. Isn't that completely counter-intuitive with the fact that rotation occurs around these bonds? I had expected the opposite effect since optical excitation is of π - π^* character, i.e. it weakens the bond order of the axles thus allowing for photo-induced rotation around these axles. Please comment. An alternative interpretation may be that rotor and stator become electronically decoupled in the excited state, and so do the normal modes of vibrations. That leads to two distinct modes and a blue-shift due a smaller reduced mass for either mode.

We agree completely with the reviewers sentiment and interpretation; this is what we meant by (p13) reproduced below:

Both modes are blue-shifted compared to GSR. We assign this to excitation localized on one axle decoupling the two halves of the motor, breaking the extended conjugation. This enhances the C=C bond order, giving rise to the observed blue-shift. In addition to the bridge C=C contribution, there is enhanced activity in the low-frequency Raman spectrum at 350, 490 and 600 cm^{-1} of the bright-state compared to the stable ground state (Figure 3c). Ring and CH modes in the 1100-1400 cm^{-1} region are also enhanced.

3.9 What is a “non-Condon” effect ? Do you mean “non-Born Oppenheimer” or an effect “beyond the Franck-Condon approximation” ? Please avoid jargon, but use common text book terminology.

By this we mean the dependence of the transition dipole moment on the nuclear coordinates. As such it is specifically a breakdown of the Franck Condon principle – the assumption that the transition moment is coordinate independent. This is somehow related to the BO approximation as suggested, but most interpret the BO as a ‘vertical transition’ which is not questioned here, so should not be invoked. “Non-Condon” is the normal usage in the literature, so we prefer to keep the current designation, but have reiterated the definition as follows (p7)

This is consistent with the coherently excited vibrational mode modulating the transition moment for emission, rather than the transition energy (which would result in out-of-phase oscillations on the red and blue edges). This suggests a coordinate dependence of the transition moment (a non-Condon effect).

3.10 Figures 18 and 19 are really difficult to read. The symmetric/anti-symmetric character of the two HF modes in fig. 18 is not obvious (arrow heads too small). Atoms superpose, but the images do not have perspective. Use different color to differentiate the rotor and stator atoms.

We thank the reviewer for pointing out this. Now we have changed the display format of molecule and background to make it clear as shown in Supplementary Figures 18 and 19. These are shown below.

Figure 18. 1554, 1540 and 53 cm^{-1} modes obtained from Gaussian DFT calculation of the stable form. The 1554 and 1540 cm^{-1} modes represent symmetric and anti-symmetric ethylenic C=C stretching of the two axles respectively. The 53 cm^{-1} mode represents a flapping-like motion of two rotors.

Figure 19. 1516, 1490 and 465 cm^{-1} modes obtained from Gaussian DFT calculation of the metastable form. The 1516 and 1490 cm^{-1} modes represent ethylenic C=C stretching of individual axle-2 and axle-1 respectively. The 465 cm^{-1} mode represents C,H out-of-plane bending motion around rotor-1 (rotor-1 is rotating around axle-1).

3.11 Proof-reading by a native speaker will allow eliminating residual typos. Some sentences are contracted, and link words like “that is” or “which is” are missing. Ex.

A native English speaker has read through the document and added links where needed.

In conclusion. The manuscript NCOMMS-22-20418 contains a number of new and highly interesting results on the ultrafast spectroscopy of this 3rd generation motor. The complementary information gained by the electronic and vibrational (not “structural”, wrong phrasing in legend of fig. 3) spectroscopies warrant to be showcased in Nature Comm.

However, not in the present form. I strongly recommend to work along the above suggestions (solvent dependent PIQY) and weaken the conclusions if needed.

We are grateful for the positive assessment and hope the new experiments address the concerns expressed.

Additional Changes made:

We have now added the FSRS data in DMSO (see Supplementary Figure 22) which also show asymmetric lineshape in early time further supporting our analysis. We have added the line in page 18 of main manuscript: Further, the FSRS data recorded in polar DMSO also show very similar spectral behavior although with a weak signal from the metastable state (see Supplementary Figure 22).

REVIEWER COMMENTS

Reviewer #1 (Remarks to the Author):

Re-review of Roy et al.

The authors have clarified and improved the manuscript following the suggestions of the reviewers. A few details should still be improved, though:

1. In Figure 3b the black histograms are often hidden underneath the red histograms. This needs to be improved.
2. The text should clarify that the computed frequencies are scaled harmonic frequencies which are used for identification of the experimentally observed signals.
3. The sentence "...adds structural dynamics data at the level of vibrational spectroscopy, with its analysis being supported by density functional theory calculations..." is misleading in the sense that the DFT calculations are static and do not inform on the underlying structural dynamics. This needs to be rephrased.

Reviewer #2 (Remarks to the Author):

I realized that the authors worked on substantial revision of the manuscript in response to the concerns and comments raised in the first round of review. In particular, I appreciate their efforts to perform additional FSRS experiments with Raman pump wavelengths as long as possible, which actually turned out to be effective in suppressing the spectral distortion. It is understandable that the authors' interpretation of the FSRS data gives a consistent picture with the population dynamics revealed by transient electronic spectroscopy, when processed by elaborate analysis with the bleach-filling method. Honestly speaking, this analysis/understanding is not the only way of interpretation. Nevertheless, the consistency between different spectroscopic data is significant enough to support that the authors' choice is one of the plausible interpretations. Considering that the target of this study is intriguing to a broad readership, the present manuscript basically deserves publication as an advanced spectroscopic report, after further improvement of the manuscript regarding points suggested below.

(1) The description on the analysis of the FSRS data is still quite confusing especially for readers outside of the field. As for the S₀ component, simply speaking, the unpumped S₀ molecule always exists and contributes to the observed signal at any delays, in addition to the component due to the excited-state molecules observed at positive delays. Thus, the S₀ component is additive in the observed signal, and hence it should be subtracted for obtaining only the component due to the excited-state molecule. However, the S₀ component is always described as a negative contribution throughout the manuscript (including the newly added main text). This confusion happens probably because the authors initially subtracted the S₀ component from the unprocessed data (directly obtained from the measurement) and used the resultant data as a starting point of their spectral analysis. In fact, all the "raw" FSRS spectra represented in the manuscript do not exhibit any noticeable band due to the

S₀ molecules at a negative delay (- 3ps). Since a portion of the molecules is excited and hence the population of the S₀ molecules is decreased at positive delays, the above-mentioned subtraction of the S₀ component becomes too much and should be compensated for. I suppose that this is a basic idea behind the bleach-filling method. However, it is somewhat difficult to understand the idea without seeing the unprocessed data and explicit explanation on the subtraction of the S₀ component. I would like to request the authors to represent the unprocessed FSRS data that are acquired directly from the measurement and explain how it is processed to obtain the “experimentally recovered FSRS” such as ESR560nm.

(2) (minor comment) The vertical axis of the FSRS graph is labelled as “FSRS signal”, but it is better to use “stimulated Raman gain” or “absorbance change”, because these physical quantities explicitly tell us their sign. Also, it is better to include a scale bar to show the magnitude of the signal.

Reviewer #3 (Remarks to the Author):

The resubmission of manuscript "Ultrafast Motion in Third Generation Photomolecular Motors" by Palas Roy et al., has very carefully taken into account all my comments and addressed the question and open issues I raised in the first report.

In particular the results with additional solvents are convincing.

The only point is 3.8. where the answer of the authors indicates that they did not quite understand my point. The excited state C=C frequency is enhanced, and "assigned to the excitation localized on one axle decoupling the two halves of the motor", breaking the extended conjugation. That's ok, and

I would suggest to continue then "This

enhances the C=C bond order on that half, giving rise to the observed blue-shift..."

But, please address also the other question: do you have indication for the bond softening (reduced bond order) on the other axle, as required for isomerisation.

Besides, the paper is really interesting and convincing for a broad audience. I am glad to recommend publication in Nature Chemistry.

We are grateful to all three reviewers for their positive assessments and their comments to improve the scientific content of the manuscript. We have now refined the paper based on the further matters noted by the reviewers (or missed by us in the first round). As before the reviewer's comments are normal typeface (marked by blue color), our replies are black italic and new text is now in orange (as it also appears in the revised manuscript, alongside red revisions from the initial review).

REVIEWER COMMENTS

Reviewer #1 (Remarks to the Author):

Re-review of Roy et al.

The authors have clarified and improved the manuscript following the suggestions of the reviewers. A few details should still be improved, though:

1. In Figure 3b the black histograms are often hidden underneath the red histograms. This needs to be improved.

Reply: A good point, We have now changed the transparency of red bars in Figure 3b in the main manuscript (here Figure 1) to see the black bars clearly.

Figure 1. DFT calculated Raman spectra of the stable (black) and metastable (red) forms of the ground state. *The computed frequencies have been scaled by a factor of 0.98.*

2. The text should clarify that the computed frequencies are scaled harmonic frequencies which are used for identification of the experimentally observed signals.

Reply: We have now added this information in the text (with the appropriate NIST citation (34)) and in the legend of Figure 3b (here in Figure 1 legend): *'The computed frequencies have been scaled by a factor of 0.98.'*^{34'}

3. The sentence "...adds structural dynamics data at the level of vibrational spectroscopy, with its analysis being supported by density functional theory calculations..." is misleading in the sense that the DFT calculations are static and do not inform on the underlying structural dynamics. This needs to be rephrased.

Reply: We have now rephrased the line in page5 as: "...adds structural dynamics data at the level of vibrational spectroscopy, with *frequency assignments* being supported by density functional theory calculations..."

Reviewer #2 (Remarks to the Author):

I realized that the authors worked on substantial revision of the manuscript in response to the concerns and comments raised in the first round of review. In particular, I appreciate their efforts to perform additional FSRS experiments with Raman pump wavelengths as long as possible, which actually turned out to be effective in suppressing the spectral distortion. It is understandable that the authors' interpretation of the FSRS data gives a consistent picture with the population dynamics revealed by transient electronic spectroscopy, when processed by elaborate analysis with the bleach-filling method. Honestly speaking, this analysis/understanding is not the only way of interpretation. Nevertheless, the consistency between different spectroscopic data is significant enough to support that the authors' choice is one of the plausible interpretations. Considering that the target of this study is intriguing to a broad readership, the present manuscript basically deserves publication as an advanced spectroscopic report, after further improvement of the manuscript regarding points suggested below.

Reply: Thank you. We believe the changes made improved the work significantly. We hope that the new text clarifies that there is space for other interpretations, although the wavelength dependence and time dependence are both (we believe) consistent with our interpretation.

(1) The description on the analysis of the FSRS data is still quite confusing especially for readers outside of the field. As for the S0 component, simply speaking, the unpumped S0 molecule always exists and contributes to the observed signal at any delays, in addition to the component due to the excited-state molecules observed at positive delays. Thus, the S0 component is additive in the observed signal, and hence it should be subtracted for obtaining only the component due to the excited-state molecule. However, the S0 component is always described as a negative contribution throughout the manuscript (including the newly added main text). This confusion happens probably because the authors initially subtracted the S0 component from the unprocessed data (directly obtained from the measurement) and used the resultant data as a starting point of their spectral analysis. In fact, all the "raw" FSRS spectra represented in the manuscript do not exhibit any noticeable band due to the S0 molecules at a negative delay (- 3ps). Since a portion of the molecules is excited and hence the population of the S0 molecules is decreased at positive delays, the above-mentioned subtraction of the S0 component becomes too much and should be compensated for. I suppose that this is a basic idea behind the bleach-filling method. However, it is somewhat difficult to understand the idea without seeing the unprocessed data and explicit explanation on the subtraction of the S0 component. I would like to request the authors to represent the unprocessed FSRS data that are acquired directly from the measurement and explain how it is processed to obtain the "experimentally recovered FSRS" such as ESR560nm.

Reply: The reviewer is absolutely correct in both the description and in stating what we have done which results in the 'negative ground state contribution', so we do need to clarify this point. To this end we have now added the unprocessed FSRS data (which includes both non-resonant solvent and pre-resonant ground state contributions and described how we processed them to generate the 'raw' excited state Raman data (ESR) as shown below in Figure 2 (and added as Supplementary Figure S23).

In the main text (p11) we state add

(see Supplementary Figure 11 for details of baseline correction procedure and Supplementary Figure S23 for details of the subtraction of the non-resonant solvent and pre-resonant GSR contributions).

And on p12

all lead to negatively signed contributions in the difference data at or near the wavenumber of GSR modes

Accordingly, we have modified the discussion in the Supplementary page 14 as follows.

Processing of the unprocessed FSRS signal at 200 fs has been shown in the Supplementary Figure 23. The unprocessed FSRS signal (grey solid line) is obtained when actinic pump, Raman pump and probe pulses are present on the sample as we detect $\log(I_{\text{Raman+Actinic+Probe}}/I_{\text{Probe}})$. On the other hand, the actinic pump and probe pulses (i.e. Raman pump OFF) generate transient absorption signal ($\log(I_{\text{Actinic+Probe}}/I_{\text{Probe}})$) as blue dotted line while the Raman pump and probe pulses (i.e. actinic OFF) provide ground state FSRS signal ($\log(I_{\text{Raman+Probe}}/I_{\text{Probe}})$) as black solid line. In our experiment, the ground state Raman signal has both solvent (large signal marked by asterisk) and motor contributions (numbered frequency). The unprocessed FSRS signal contains information on the excited state Raman along with the ground state Raman, transient absorption (TA) and nonlinear background. Therefore the ground state Raman and TA contributions have been subtracted from the unprocessed FSRS data. The resultant difference spectra (say, raw excited state Raman signal ($\text{ESR}_{\lambda_{\text{Raman}}}$)) are shown as red solid line. Same procedure is followed to produce raw ESR data at different time delays which are shown in Supplementary Figures 9a and 10a.

And on Supplementary p31

Figure 2. Procedure to generate raw excited state Raman data (ESR) from the unprocessed FSRS signal of 3GMph in ACN at 200 fs utilising Raman pump at a) 560 and b) 650 nm. The transient absorption (blue dotted line, B) and ground state Raman (black solid line, C) contributions have been subtracted from its unprocessed FSRS data with actinic ON (grey solid line, A) to generate the ESR_{560} and ESR_{650} (red solid line).

(2) (minor comment) The vertical axis of the FSRS graph is labelled as “FSRS signal”, but it is better to use “stimulated Raman gain” or “absorbance change”, because these physical quantities explicitly tell us their sign. Also, it is better to include a scale bar to show the magnitude of the signal.

Reply: We have now replaced the ‘FSRS signal’ by ‘Stimulated Raman Gain’ in the Y axis of all the Figures (Main Figures 3a, 3c, 3d, 4c and Supplementary Figure 11). We prefer not to add a scale bar to the stack plots. They are in the conventional format for FSRS and would rather keep it in this style.

It is a more convenient way to compare and observe the time evolution. We agree that the absolute magnitude of the signal can be useful information, so on Figure S23 we add a second vertical axis to show the optical density change observed.

Reviewer #3 (Remarks to the Author):

The resubmission of manuscript "Ultrafast Motion in Third Generation Photomolecular Motors" by Palas Roy et al., has very carefully taken into account all my comments and addressed the question and open issues I raised in the first report.

In particular the results with additional solvents are convincing.

Reply: *Thank you. Again, we believe these additions materially improved the manuscript.*

The only point is 3.8. where the answer of the authors indicates that they did not quite understand my point. The excited state C=C frequency is enhanced, and "assigned to the excitation localized on one axle decoupling the two halves of the motor", breaking the extended conjugation. That's ok, and I would suggest to continue then "This enhances the C=C bond order on that half, giving rise to the observed blue-shift..."

Reply: *Agreed. We have now modified the line in the main manuscript page14 as: 'This enhances the C=C bond order **on that half**, giving rise to the observed blue-shift.'*

But, please address also the other question: do you have indication for the bond softening (reduced bond order) on the other axle, as required for isomerisation.

Reply: *Yes, the FSRS data of the dark state (prior to isomerization) in main Figure 3e shows that both axle stretching frequency modes (around 1550 cm^{-1}) are red-shifted indicating 'softening' of the axle modes during isomerisation.*

To highlight this we add on p14

(i.e. the C=C bond strength decreases during excited state evolution to the dark state and prior to formation of the ground state product)

Besides, the paper is really interesting and convincing for a broad audience. I am glad to recommend publication in Nature Chemistry.

Reply: *Thank you.*

In summary, in the preceding we have endeavoured to address all the reviewers' comments on the revised version of the manuscript.

REVIEWERS' COMMENTS

Reviewer #2 (Remarks to the Author):

Going through the revised manuscript, I realized that the authors and this reviewer have been able to share a common understanding about the concerns that I raised in the previous rounds of review. I have made sure that the authors addressed all the points and improved the text properly, allowing a broad range of readers to understand correctly and smoothly what the authors meant to present with their spectroscopy on the intriguing target. Now I feel that the current form of the manuscript satisfies the criteria, so that I am happy to recommend its publication in Nature Comm.